# Mixed formulation for an easy and robust numerical computation of sorptivity

Laurent Lassabatere[1], Pierre-Emmanuel Peyneau[2], Deniz Yilmaz[3], Joseph Pollacco[4],
Jesús Fernández-Gálvez[5], Borja Latorre[6], David Moret-Fernández[6], Simone Di Prima[7],
Mehdi Rahmati[8,9], Ryan D. Stewart[10], Majdi Abou Najm[11], Claude Hammecker[12], and
Rafael Angulo-Jaramillo[1]

[1]Univ Lyon, Université Claude Bernard Lyon 1, CNRS, ENTPE, UMR5023 LEHNA, F-69518, Vaulx-en-Velin, France
[2]GERS-LEE, Univ Gustave Eiffel, IFSTTAR, F-44344 Bouguenais, France
[3]Civil Engineering Department, Engineering Faculty, Munzur University, Tunceli, Turkey
[4]Manaaki Whenua - Landcare Research, 7640 Lincoln, New Zealand
[5]Department of Regional Geographic Analysis and Physical Geography, University of Granada, 18071 Granada, Spain
[6]Departamento de Suelo y Agua, Estación Experimental de Aula Dei, Consejo Superior de Investigaciones Científicas
(CSIC), PO Box 13034, 50080 Zaragoza, Spain
[7]Agricultural Department, University of Sassari, Viale Italia, 39, 07100 Sassari, Italy
[8]Department of Soil Science and Engineering, Faculty of Agriculture, University of Maragheh, Maragheh, Iran
[9]Forschungszentrum Jülich GmbH, Institute of Bio- and Geosciences: Agrosphere (IBG-3), Jülich, Germany
[10]School of Plant and Environmental Sciences, Virginia Polytechnic Institute and State University, Blacksburg, VA, United
States
[11]Department of Land, Air and Water Resources, University of California, Davis, CA 95616, United States
[12]University of Montpellier,UMR LISAH, IRD, Montpellier, France

**Correspondence:** Laurent Lassabatere (laurent.lassabatere@entpe.fr)

**Abstract.**

Sorptivity is one of the most important parameters for the quantification of water infiltration into soils. Parlange (1975) proposed a specific formulation to derive sorptivity as a function of the soil water retention and hydraulic conductivity functions, as well as initial and final soil water contents. However, this formulation requires the integration of a function involving the hydraulic diffusivity, which may be undefined or present numerical difficulties that cause numerical misestimations. In this study, we propose a mixed formulation that scales sorptivity and splits the integrals into two parts: the first term involves the scaled degree of saturation while the second involves the scaled water pressure head. The new mixed formulation is shown to be robust and well-suited to any type of hydraulic functions - even with infinite hydraulic diffusivity or positive air-entry water pressure heads - and any boundary condition, including infinite initial water pressure head, $h \rightarrow -\infty$. Lastly, we show the benefits of using the proposed formulation for modeling water into soil with analytical models that use sorptivity.

## 1 Introduction

Soil sorptivity represents the capacity of soil to absorb water by capillarity (Cook and Minasny, 2011). The accurate estimation of soil sorptivity is crucial for the modeling of water infiltration into soils and the hydraulic characterization of soils (Angulo-

Jaramillo et al., 2016; Stewart and Abou Najm, 2018). Several models and methods make use of this variable, such as in the Beerkan Estimation of Soil Transfer parameters (BEST) methods (Lassabatere et al., 2006; Yilmaz et al., 2010; Bagarello et al., 2014a; Lassabatere et al., 2019; Angulo-Jaramillo et al., 2019) and related simplified Beerkan approaches (Bagarello et al., 2014b; Di Prima et al., 2020; Yilmaz, 2021). Sorptivity is also required for the computation of several hydraulic parameters, like the macroscopic capillary length (Bouwer, 1964; White and Sully, 1987).

The squared sorptivity is related to the flux concentration function, $F(\theta)$, as follows (Philip and Knight, 1974):

$$S^2(\theta_0, \theta_1) = 2 \int_{\theta_0}^{\theta_1} \frac{(\theta - \theta_0)}{F(\theta)} D(\theta) \, d\theta \tag{1}$$

where $D(\theta) = K(\theta) \, dh/d\theta$ is the hydraulic diffusivity function, $\theta_0$ and $\theta_1$ stand for the initial and final water contents. In the context of water infiltration into soils, the initial water content refers to the water content along the soil profile before water infiltration, and the final water content corresponds to that imposed at the soil surface (at the water source). Several studies have investigated the definition of the flux-concentration functions, depending on the type of soils (Angulo-Jaramillo et al., 2016, Table 1, pp. 33). Ross et al. (1996) suggested the use of the approximation proposed by Parlange (1975) for most soils leading to two main forms of sorptivity, a diffusivity form, $S_D$, and a conductivity form, $S_K$, both summarized below:

$$F(\theta) = \frac{2(\theta - \theta_0)}{(\theta_1 + \theta - 2\theta_0)} \tag{2}$$

$$S_{\mathrm{D}}^2(\theta_0, \theta_1) = \int_{\theta_0}^{\theta_1} (\theta_1 + \theta - 2\theta_0) D(\theta) \, d\theta \tag{3}$$

$$S_{\mathrm{D}}(\theta_0, \theta_1) = \sqrt{\int_{\theta_0}^{\theta_1} (\theta_1 + \theta - 2\theta_0) D(\theta) \, d\theta} \tag{4}$$

$$S_{\mathrm{K}}(h_0, h_1) = \sqrt{\int_{h_0}^{h_1} (\theta_1 + \theta(h) - 2\theta_0) K(h) \, dh} \tag{5}$$

where the initial and the final values of the water pressure heads, $h_0$ and $h_1$, correspond to the water contents $\theta_0 = \theta(h_0)$ and $\theta_1 = \theta(h_1)$.

The two forms, the diffusivity form, $S_{\mathrm{D}}$, and the conductivity form, $S_{\mathrm{K}}$, each have their own shortcomings. For certain hydraulic models, $D(\theta)$ tends towards infinity when $\theta_0 \to \theta_s$, making it difficult to compute the right-hand side of Eq. (4). Moreover, when the surface water pressure head exceeds the air-entry water pressure head, $S_{\mathrm{D}}$ misses the saturated part of sorptivity, $\int_{h_a}^{h_1} (\theta(h_1) + \theta(h) - 2\theta(h_0)) K(h) \, dh$ (Ross et al., 1996). The conductivity form $S_{\mathrm{K}}$ must be used when it is

necessary to account for the two parts of sorptivity, i.e., the unsaturated and saturated parts, as indicated by the following relationship (Lassabatere et al., 2021):

$$S_{\mathrm{K}}\left(h_0, h_1 \geq h_{\mathrm{a}}\right) = \sqrt{S_{\mathrm{D}}^2\left(\theta_0, \theta_{\mathrm{s}}\right) + 2\left(\theta_{\mathrm{s}} - \theta_0\right) K_{\mathrm{s}}\left(h_1 - h_{\mathrm{a}}\right)}$$
$$= \sqrt{S_{\mathrm{D}}^2\left(\theta\left(h_0\right), \theta\left(h_1\right)\right) + 2\left(\theta\left(h_1\right) - \theta\left(h_0\right)\right) K_{\mathrm{s}}\left(h_1 - h_{\mathrm{a}}\right)} \tag{6}$$

We can thus conclude that the conductivity form, $S_{\mathrm{K}}$, is the more general equation. However, $S_{\mathrm{K}}$ can also be difficult to handle when the initial conditions are very dry. In particular, for very dry initial conditions, the initial water pressure head

corresponding to $\theta_r$ corresponds to $h_0 \to -\infty$. Then, the calculation of $S_{\mathrm{K}}$ requires the evaluation of an integral that involves an infinite lower bound: $\int_{-\infty}^{h_1}\left(\theta\left(h_1\right) + \theta\left(h\right) - 2\,\theta_r\right) K\left(h\right) dh$.

In this study, we propose a new mixed formulation that overcomes these problems. We compare it to the approaches commonly used to compute sorptivity, i.e., Eq. (4) and Eq. (5). The proposed mixed formulation automatically accounts for the saturated and unsaturated parts of sorptivity. It also allows for easy computation under any initial condition, including the

extreme case of an initial water content equal to the residual water content, $\theta_0 = \theta_r$ (corresponding to a negative initial water pressure head that tends towards infinity, $h_0 \to -\infty$) and a final water pressure head higher than the air-entry water pressure head, $h_1 \geq h_a$, even including positive values. In addition to proposing a new robust formulation for the computation of sorptivity, we aim to demonstrate the following points: (i) the computation of sorptivity with classic approaches may be challenging depending on the hydraulic models chosen for describing the water retention (WR) and hydraulic conductivity (HC) functions,

(ii) the proposed mixed formulation is an ideal estimator for sorptivity and performs well at all times, (iii) the usual methods, based on the use of $S_{\mathrm{D}}$ (Eq. 4) or $S_{\mathrm{K}}$ (Eq. 5), do not necessarily provide accurate estimations of the nominal sorptivity, (iv) those misestimations of sorptivity may have substantial impacts on the prediction of water infiltration into soils, when inserting sorptivity into analytical models.

The paper is organized as follows. The theory section presents the proposed mixed formulation. Next, the paper analyzes

the precision of the mixed formulation by comparing it with the exact analytical formulation for the case of the maximum sorptivity, $S\left(-\infty, 0\right) = \sqrt{\int_{-\infty}^{0}\left(\theta_s + \theta\left(h\right) - 2\,\theta_r\right) K\left(h\right) dh}$. The maximum sorptivity, $S\left(-\infty, 0\right)$, encompasses the two types of problems, i.e., infinite negative initial water pressure head, and infinite diffusivity function close to water saturation $\theta \to \theta_s$, and also omission of the saturated part of sorptivity when $h_1 > h_a$ by regular approaches. We considered three commonly used hydraulic models, for which Lassabatere et al. (2021) proposed analytical formulations for $S\left(-\infty, 0\right)$: Brooks and Corey (BC),

van Genuchten - Burdine (vGB), and van Genuchten - Mualem (vGM). The second part of the paper compares the accuracy of the mixed formulation with the current strategies for the same three hydraulic models plus the Kosugi (KG) model and demonstrates the risk of serious misestimations with prior approaches. By presenting a new formulation that is applicable to any types of conditions, this paper completes the study of Lassabatere et al. (2021), who proposed a scaling procedure for the approximation of $S_{\mathrm{K}}\left(h_0, h_1 = 0\right)$ with the condition of null water pressure head at surface, i.e., $h_1 = 0$. Lastly, we show

how using the proposed approach can improve the accuracy of sorptivity estimation and, consequently, the modeling of water infiltration into soils with analytical models that make use of sorptivity.

## 2 Theory

### 2.1 Proposed new mixed formulation for computing sorptivity

To build the mixed formulation, $S_\mathrm{M}$, we start with the conductivity form of sorptivity, $S_\mathrm{K}$, since it includes both unsaturated and saturated parts. Then, we define an intermediate water pressure head between the initial and final water pressure heads, $h_\mathrm{c} \in [h_0, h_1]$, smaller than the air entry pressure, $h_\mathrm{c} < h_a \leq 0$, and we split the integral into two separate parts as follows:

$$
S_\mathrm{M}(h_0, h_1) = S_\mathrm{K}(h_0, h_1) = \sqrt{\int_{h_0}^{h_1} (\theta(h_1) + \theta(h) - 2\theta(h_0)) K(h) \, dh}
$$

$$
= \sqrt{\int_{h_0}^{h_\mathrm{c}} (\theta(h_1) + \theta(h) - 2\theta(h_0)) K(h) \, dh + \int_{h_\mathrm{c}}^{h_1} (\theta(h_1) + \theta(h) - 2\theta(h_0)) K(h) \, dh}
$$

$$
= \sqrt{\int_{\theta(h_0)}^{\theta(h_\mathrm{c})} (\theta(h_1) + \theta - 2\theta(h_0)) D(\theta) \, d\theta + \int_{h_\mathrm{c}}^{h_1} (\theta(h_1) + \theta(h) - 2\theta(h_0)) K(h) \, dh} \tag{7}
$$

In Eq. (7), the integral $\int_{h_0}^{h_\mathrm{c}} (\theta(h_1) + \theta(h) - 2\theta(h_0)) K(h) \, dh$ is transformed into $\int_{\theta(h_0)}^{\theta(h_\mathrm{c})} (\theta(h_1) + \theta - 2\theta(h_0)) D(\theta) \, d\theta$ thanks to the change of variable $h \to \theta$. This operation requires that the function $\theta(h)$ is bijective over the whole interval $[h_0, h_\mathrm{c}]$, which is valid so long as $h_\mathrm{c} < h_a$. The mixed formulation, $S_\mathrm{M}$, may be written alternatively as follows:

$$
S_\mathrm{M}(h_0, h_1) = \sqrt{\underbrace{\int_{\theta_0}^{\theta_\mathrm{c}} (\theta_1 + \theta - 2\theta_0) D(\theta) \, d\theta}_{A} + \underbrace{\int_{h_\mathrm{c}}^{h_1} (\theta_1 + \theta(h) - 2\theta_0) K(h) \, dh}_{B}} \tag{8}
$$

where $\theta_\mathrm{c} = \theta(h_\mathrm{c})$, $\theta_1 = \theta(h_1)$, and $\theta_0 = \theta(h_0)$. The constraint $h_\mathrm{c} < h_\mathrm{a}$ ensures that the computation of A in Eq. (8) avoids the challenging integration of infinite diffusivity close to saturation, $D(\theta_\mathrm{s}) = +\infty$, since $\theta_\mathrm{c} < \theta_\mathrm{s}$. In addition, $h_\mathrm{c}$ is bounded to finite value to avoid integration over infinite intervals for part B. Then, the two integrals involved in Eq. (8), A and B only involve bounded functions over finite intervals, ensuring an easy numerical computation. An illustration of the procedure is depicted in Figure (1).

Next, we scale sorptivity to separate the respective contributions of scale and shape parameters, as suggested by Lassabatere et al. (2021). We consider the following scaling relationships for hydraulic variables and sorptivity:

$$
\begin{cases}
S_\mathrm{e} = \dfrac{\theta - \theta_\mathrm{r}}{\theta_\mathrm{s} - \theta_\mathrm{r}} \\[2mm]
h^* = \dfrac{h}{|h_\mathrm{g}|} \\[2mm]
K_\mathrm{r} = \dfrac{K}{K_\mathrm{s}}
\end{cases} \tag{9}
$$

where $S_\mathrm{e}$ is the saturation degree, $h^*$ is the scaled water pressure head, $K_\mathrm{r}$ is the relative hydraulic conductivity, $\theta_\mathrm{r}$ and $\theta_\mathrm{s}$ are the residual and the saturated water contents, $h_\mathrm{g}$ is the scale parameter for the water pressure head, and $K_\mathrm{s}$ is the saturated

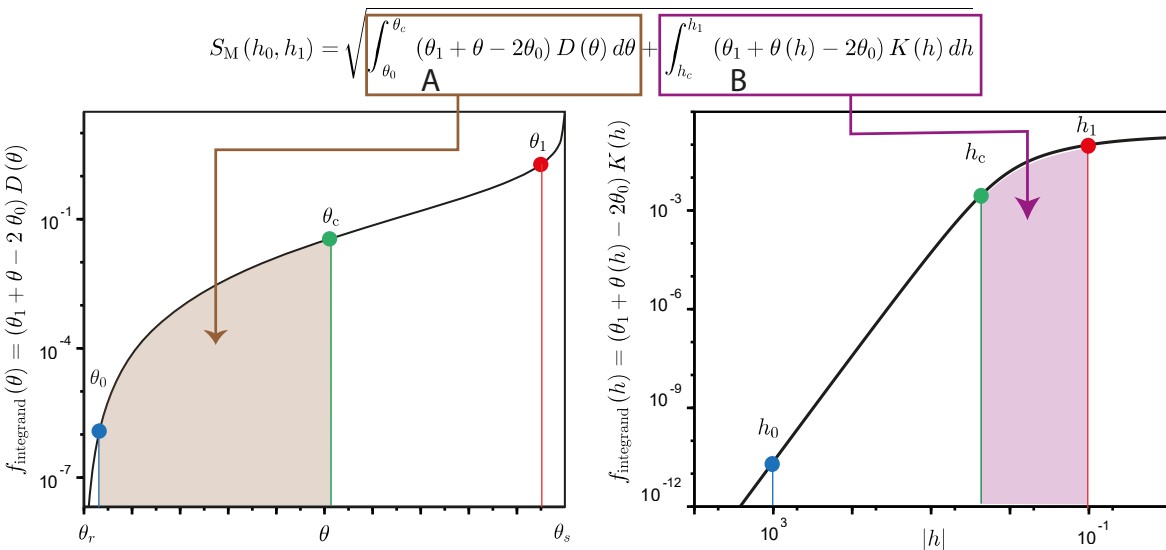

$$S_{\mathrm{M}}(h_0, h_1) = \sqrt{\underbrace{\int_{\theta_0}^{\theta_c} (\theta_1 + \theta - 2\theta_0)\, D(\theta)\, d\theta}_{\mathrm{A}} + \underbrace{\int_{h_c}^{h_1} (\theta_1 + \theta(h) - 2\theta_0)\, K(h)\, dh}_{\mathrm{B}}}$$

**Figure 1.** Concept of the mixed formulation, $S_{\mathrm{M}}(h_0, h_1)$: the integration of $\int_{h_0}^{h_1}(\theta_1 + \theta(h) - 2\theta_0)\, K(h)\, dh$ $\left(= S_{\mathrm{M}}^2(h_0, h_1)\right)$ is converted into the sum of the integration of two bounded functions over bounded intervals, $\int_{\theta_0}^{\theta_c}(\theta_1 + \theta - 2\theta_0)\, D(\theta)\, d\theta$ and $\int_{h_c}^{h_1}(\theta_1 + \theta(h) - 2\theta_0)\, K(h)\, dh$. Note that the data are depicted with log-scale for clarity, but the integration performs directly on the integrands instead of their log-scaled counterparts. Consequently, the integrals do not correspond to the areas below the curves, conversely to the case of linear scales. Illustrative case of the computation of sorptivity, $S_{\mathrm{M}}(h_0, h_1)$, for a synthetic loamy soil with an initial water pressure head of $-10^3$ mm and a final water pressure head of $-10^{-1}$ mm.

hydraulic conductivity. The application of scaling relationships of Eq. (9) to the dimensional sorptivity expressions leads to the following equation (Ross et al., 1996):

$$S = \sqrt{|h_{\mathrm{g}}| K_{\mathrm{s}} (\theta_{\mathrm{s}} - \theta_{\mathrm{r}})}\, S^* \tag{10}$$

where $S$ and $S^*$ are respectively the dimensional and the scaled sorptivities. The application of the scaling equations Eqs. (9-10) to the mixed formulation $S_{\mathrm{M}}$, defined by Eq. (7), leads to the final expression proposed in our study:

$$\begin{cases} S_{\mathrm{M}}(h_0, h_1) = \sqrt{|h_{\mathrm{g}}| K_{\mathrm{s}} (\theta_{\mathrm{s}} - \theta_{\mathrm{r}})}\, S_{\mathrm{M}}^*(h_0^*, h_1^*) \\ S_{\mathrm{M}}^*(h_0^*, h_1^*) = \sqrt{\int_{S_{\mathrm{e}}(h_0^*)}^{S_{\mathrm{e}}(h_c^*)} (S_{\mathrm{e}}(h_1^*) + S_{\mathrm{e}} - 2 S_{\mathrm{e}}(h_0^*))\, D^*(S_{\mathrm{e}})\, dS_{\mathrm{e}} + \int_{h_c^*}^{h_1^*} (S_{\mathrm{e}}(h_1^*) + S_{\mathrm{e}}(h^*) - 2 S_{\mathrm{e}}(h_0^*))\, K_{\mathrm{r}}(h^*)\, dh^*} \end{cases} \tag{11}$$

where $S_{\mathrm{M}}^*(h_0^*, h_1^*)$ is the scaled version of the proposed mixed formulation, $S_{\mathrm{M}}(h_0, h_1)$, with $h_0^* = h_0/|h_{\mathrm{g}}|$ and $h_1^* = h_1/|h_{\mathrm{g}}|$. Eq. (11) can be demonstrated by changing the integration variable $\theta \to S_{\mathrm{e}}$ in the first and $h \to h^*$ in the second integral

of Eq. (7). Eq. (11) cannot be evaluated with coding software that do not allow infinite values, $h_0^* = -\infty$. We then replace the input $h_0$ (that may be infinite) with $S_{e,0} = S_e(h_0^*)$, which always remains bounded in the final expression of the mixed

formulation $S_M^*$:

$$S_M^* \left(S_{e,0}, h_1^*\right) = \sqrt{\int\limits_{S_{e,0}}^{S_e(h_c^*)} \left(S_e\left(h_1^*\right) + S_e - 2 S_{e,0}\right) D^*\left(S_e\right) dS_e + \int\limits_{h_c^*}^{h_1^*} \left(S_e\left(h_1^*\right) + S_e\left(h^*\right) - 2 S_{e,0}\right) K_r\left(h^*\right) dh^*} \tag{12}$$

Several options exist for the choice of the intermediate water pressure head $h_c^*$ and intermediate saturation degree $S_{e,c} = S_e\left(h_c^*\right)$. In this study, our preferred option is to set the intermediate saturation degree as the average between the initial and the final saturation degrees, $S_{e,c} = \left(S_{e,0} + S_{e,1}\right)/2$. However, under certain circumstances (e.g., for soils with gradual water retention functions, see Results section), the value of $h^*(S_{e,c})$ may reach very large values, leading to numerical instabilities. Therefore, we use the following criteria to ensure that $h^*(S_{e,c})$ remains finite:

$$\begin{cases} h_c^* = -\min\left(\left|h^*\left(\frac{S_{e,0}+S_{e,1}}{2}\right)\right|, 10^z\right) & z \in \mathbb{Z} \\ S_{e,c} = S_e\left(h_c^*\right) \end{cases} \tag{13}$$

When necessary, the value of $z$ is varied until the two integrals in $S_M^*$ (Eq. 12) converge. In most cases, $z \in \{-2, -1, 0, 1, 2\}$ ensures convergence regardless of soil type and situation. Note that for hydraulic models with non-null water entry pressure head $h_a < 0$, $z$ should be fixed with $z \geq 0$ so as to ensure $-10^z \leq -1$ and thus $h_c^* \leq h_a^* = -1$. This condition is necessary to ensure the bijectivity of the function $S_e\left(h^*\right)$ over the interval $[h_0^*, h_c^*]$, which is required for the use of Eq. (12).

In the following, the mixed formulation $S_M^*$, Eqs. (12-13) will be compared to several strategies previously proposed in the literature to cope with situations of numerical indeterminacy, e.g., at saturation $\theta_1 = \theta_s$ (or, $S_{e,1} = 1$) for a null water pressure head at surface, $h_1 = 0$, and for very dry initial conditions $\theta_0 \to \theta_r$ (or, $h_0^* \to -\infty$).

## 2.2 Usual methods for computing sorptivity based on $S_D$ and $S_K$

### 2.2.1 Computing sorptivity with $S_K$ for very dry initial conditions, $h_0 \to -\infty$

Regarding the computation of sorptivity for very dry initial conditions with $S_K$, one of the strategies found in the literature applies the regular definitions of sorptivity (Parlange, 1975), Eq. (5), to the case of very low values of $h_0$. Such an approach was used by Di Prima et al. (2020) for the estimation of $S_K\left(h_0, h_1\right)$ for very dry soils. In this case, the maximum sorptivity, $S\left(-\infty, h_1\right)$, is approached as follows:

$$S\left(-\infty, h_1\right) = \sqrt{\int\limits_{-\infty}^{h_1} \left(\theta_1 + \theta\left(h\right) - 2\,\theta_r\right) K\left(h\right) dh}$$

$$= \lim_{h_0 \to -\infty} \sqrt{\int\limits_{h_0}^{h_1} \left(\theta_1 + \theta\left(h\right) - 2\theta\left(\underline{h_0}\right)\right) K\left(h\right) dh}$$

$$= \lim_{h_0 \to -\infty} S_K\left(\underline{h_0}, h_1\right) \tag{14}$$

Note that, for the sake of clarity, the underline in Eq. (14) shows the variables that are varied. Note also that the preceding equations are valid since $\sqrt{\lim f(x)} = \lim \sqrt{f(x)}$, with $y = \sqrt{x}$ defining a continuous function. In practical applications of this method, $S_{\mathrm{K}}(h_0, h_1)$ is computed for decreasing values of $h_0$ until reaching stabilization (Fig. 2a). The last value obtained in this way is considered to represent the sorptivity at extremely dry conditions, i.e., $S(-\infty, h_1)$. This option is quite practical since it requires the user to only code the regular function $S_{\mathrm{K}}$ before applying it to very negative values of $h_0$.

We propose an alternative procedure that employs a specific integrand to compute the same limit (Fig. 2b). In this case, the water content $\theta_0$ is set equal to $\theta_r$ in the integrand so as to correspond to the targeted initial conditions $\theta(h_0 = -\infty) = \theta_r$:

$$
\begin{aligned}
S(-\infty, h_1) &= \sqrt{\int_{-\infty}^{h_1} (\theta_1 + \theta(h) - 2\,\theta_r)\, K(h)\, dh} \\[2mm]
&= \lim_{h_0 \to -\infty} \sqrt{\int_{\underline{h_0}}^{h_1} (\theta_1 + \theta(h) - 2\,\theta_r)\, K(h)\, dh} \\[2mm]
&= \lim_{h_0 \to -\infty} S_{\mathrm{K-V2}}\left(\underline{h_0}, h_1\right)
\end{aligned}
\tag{15}
$$

with the specific function $S_{\mathrm{K-V2}}$ defined as follows:

$$
S_{\mathrm{K-V2}}(h_0, h_1) = \sqrt{\int_{h_0}^{h_1} (\theta_1 + \theta(h) - 2\,\theta_r)\, K(h)\, dh}
\tag{16}
$$

In comparison to $S_{\mathrm{K}}$, the water content $\theta_0$ is replaced with $\theta_{\mathrm{r}}$ in the integrand for $S_{\mathrm{K-V2}}$. We expect this modification to improve numerical convergence towards the lower integration limit, since $S_{\mathrm{K-V2}}$ directly integrates the right integrand (Fig. 2b). Conversely, $S_{\mathrm{K}}$ integrates a distinct integrand, i.e., $(\theta_1 + \theta(h) - 2\theta(h_0))\, K(h) \neq (\theta_1 + \theta(h) - 2\theta_{\mathrm{r}})\, K(h)$, thus involving an additional source of error (Fig. 2a). Briefly, $S_{\mathrm{K}}$ combines the error due to the integral bound $h_0 > -\infty$ and the difference between its integrand and the targeted integrand. Note that the function $S_{\mathrm{K-V2}}$ should be restricted to the evaluation of $S(-\infty, h_1)$ and never used for the computation of other cases, i.e., $S(h_0 \neq -\infty, h_1)$, since the water content in the integrand is fixed at $\theta_{\mathrm{r}}$, which corresponds exclusively to the case of $h_0 = -\infty$.

The application of the scaling, Eqs. (9-10), to the preceding definitions, Eqs. (14-15), leads to scaled versions of those equations, which will be used in the computations below:

$$
\begin{aligned}
S^*(-\infty, h_1^*) &= \lim_{h_0^* \to -\infty} \sqrt{\int_{\underline{h_0^*}}^{h_1^*} \left(S_{\mathrm{e},1} + S_{\mathrm{e}}(h^*) - 2\, S_{\mathrm{e}}\left(\underline{h_0^*}\right)\right) K_{\mathrm{r}}(h^*)\, dh^*} \\[2mm]
&= \lim_{h_0^* \to -\infty} S_{\mathrm{K}}^*\left(\underline{h_0^*}, h_1^*\right)
\end{aligned}
\tag{17}
$$

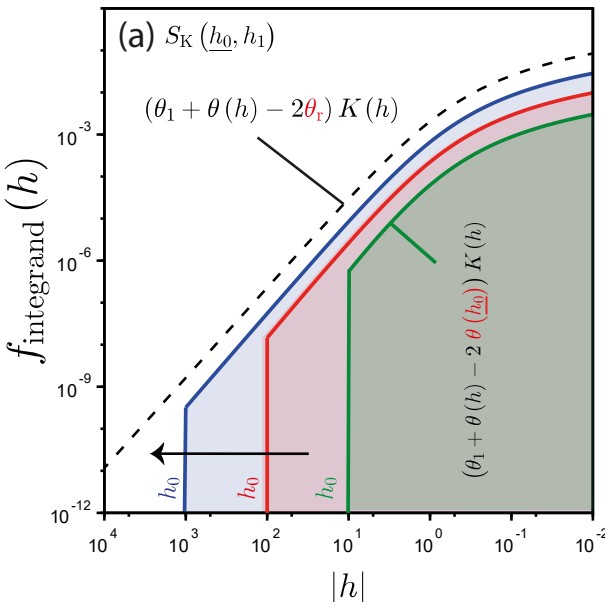 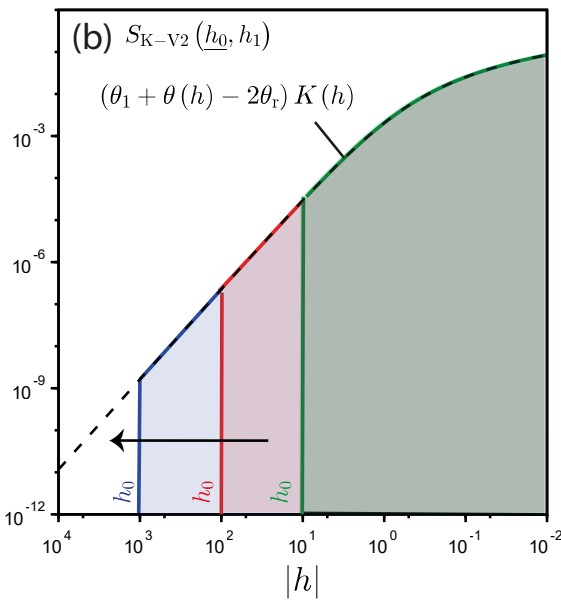

**Figure 2.** Illustration of the regular strategies for the estimation of the limits for the case of very dry conditions with the estimation of $S(-\infty, h_1)$ using either $S_\mathrm{K}(h_0, h_1)$, Eq. 5, (a) or $S_{\mathrm{K-V2}}(h_0, h_1)$, Eq. 16, (b). The integration proceeds from the given value of $h_0$ to the right, $h \geq h_0$, corresponding to the solid zones, and the value of $h_0$ is lowered step by step to reach the limit $S(-\infty, h_1)$ (see the arrow and the extension of the solid zones). In the equations, the variables that are varied to reach the targeted limits are underlined, and figures are zoomed in in the vicinity of the limits. The differences in equations between the target and the integrated integrands are in red. Illustrative case of the computation of the limits of $S_\mathrm{K}(h_0, h_1)$ or $S_{\mathrm{K-V2}}(h_0, h_1)$ for a synthetic loamy soil with a final water pressure head of $h_1 = -1$ mm; successive values are considered for the initial water pressure heads, $h_0 \in \{-10^1, -10^2, -10^3\}$ mm.

$$
\begin{aligned}
S^*(-\infty, h_1^*) &= \lim_{h_0^* \to -\infty} \sqrt{\int_{h_0^*}^{h_1^*} (S_{\mathrm{e},1} + S_\mathrm{e}(h^*)) K_\mathrm{r}(h^*) \, dh^*} \\
&= \lim_{h_0^* \to -\infty} S_{\mathrm{K-V2}}^*\left(\underline{h_0^*}, h_1^*\right)
\end{aligned}
\tag{18}
$$

with $S_\mathrm{K}^*$ and $S_{\mathrm{K-V2}}^*$, the scaled versions of $S_\mathrm{K}$ and $S_{\mathrm{K-V2}}$, defined as follows:

$$
\begin{cases}
S_\mathrm{K}^*(h_0^*, h_1^*) = \sqrt{\int_{h_0^*}^{h_1^*} (S_{\mathrm{e},1} + S_\mathrm{e}(h^*) - 2S_\mathrm{e}(h_0^*)) K_\mathrm{r}(h^*) \, dh^*} \\
S_{\mathrm{K-V2}}^*(h_0^*, h_1^*) = \sqrt{\int_{h_0^*}^{h_1^*} (S_{\mathrm{e},1} + S_\mathrm{e}(h^*)) K_\mathrm{r}(h^*) \, dh^*}
\end{cases}
\tag{19}
$$

The derivation of these equations involved scaling Eq. (9) and Eq. (10) along with the change of variable $\theta \to S_\mathrm{e}$.

### 2.2.2 Computing sorptivity with $S_\mathrm{D}$ for null water pressure head at surface ($h_1 = 0$)

A similar approach is often used with the $S_\mathrm{D}$ formulation to avoid numerical indeterminacy close to saturation. The first option considers $S_\mathrm{D}(\theta_0, \theta_1)$ with $\theta_1 \to \theta_s$ as suggested, for instance, by Fernández-Gálvez et al. (2019):

$$
\begin{aligned}
S_\mathrm{u}(\theta_0, \theta_\mathrm{s}) &= \sqrt{\int_{\theta_0}^{\theta_\mathrm{s}} (\theta_\mathrm{s} + \theta - 2\,\theta_0)\, D(\theta)\, d\theta} \\
&= \lim_{\theta_1 \to \theta_\mathrm{s}} \sqrt{\int_{\theta_0}^{\theta_1} (\underline{\theta_1} + \theta - 2\,\theta_0)\, D(\theta)\, d\theta} \\
&= \lim_{\theta_1 \to \theta_\mathrm{s}} S_\mathrm{D}(\theta_0, \underline{\theta_1})
\end{aligned}
\tag{20}
$$

Note that with this method, we can only account for the unsaturated part of sorptivity, $S_\mathrm{u}(\theta_0, \theta_\mathrm{s}) = \sqrt{\int_{\theta_0}^{\theta_\mathrm{s}} (\theta_\mathrm{s} + \theta - 2\,\theta_0)\, D(\theta)\, d\theta}$, and we systematically miss the saturated portion of sorptivity $2\,(\theta_\mathrm{s} - \theta_0)\, K_\mathrm{s}(h_1 - h_\mathrm{a})$, as mentioned in section 1. The total sorptivity corresponds to the sum of its two components (see Eq. 6): $S(h_0, h_1 > h_a) = \sqrt{S_\mathrm{u}^2(\theta_0, \theta_\mathrm{s}) + 2\,(\theta_\mathrm{s} - \theta_0)\, K_s(h_1 - h_a)}$. The subscript "u" in $S_\mathrm{u}(\theta_0, \theta_\mathrm{s})$ stands for "unsaturated" and serves as a reminder of that limitation (Ross et al., 1996). This point will be further illustrated and discussed in the Results section.

The integrand specified by $S_\mathrm{D}$ corresponds to $(\theta_1 + \theta(h) - 2\,\theta_0)\, D(\theta)$. Consequently, $S_\mathrm{D}$ combines the error due to the discrepancy between the integrated and the targeted integrands with the error resulting from the restriction of the integration to $[\theta_0, \theta_1]$ instead of $[\theta_0, \theta_\mathrm{s}]$ (Fig. 3a, $S_\mathrm{D}$). To correct this problem, we define a different estimator, $S_{\mathrm{D-V2}}$, to integrate directly the targeted integrand, $(\theta_\mathrm{s} + \theta(h) - 2\,\theta_0)\, D(\theta)$ (Fig. 3b, $S_{\mathrm{D-V2}}$), and the following developments emerge:

$$
\begin{aligned}
S_\mathrm{u}(\theta_0, \theta_\mathrm{s}) &= \sqrt{\int_{\theta_0}^{\theta_\mathrm{s}} (\theta_\mathrm{s} + \theta - 2\,\theta_0)\, D(\theta)\, d\theta} \\
&= \lim_{\theta_1 \to \theta_\mathrm{s}} \sqrt{\int_{\theta_0}^{\theta_1} (\theta_\mathrm{s} + \theta - 2\,\theta_0)\, D(\theta)\, d\theta} \\
&= \lim_{\theta_1 \to \theta_\mathrm{s}} S_{\mathrm{D-V2}}(\theta_0, \underline{\theta_1})
\end{aligned}
\tag{21}
$$

with the function $S_{\mathrm{D-V2}}$ defined as follows:

$$
S_{\mathrm{D-V2}}(\theta_0, \theta_1) = \sqrt{\int_{\theta_0}^{\theta_1} (\theta_\mathrm{s} + \theta - 2\,\theta_0)\, D(\theta)\, d\theta}
\tag{22}
$$

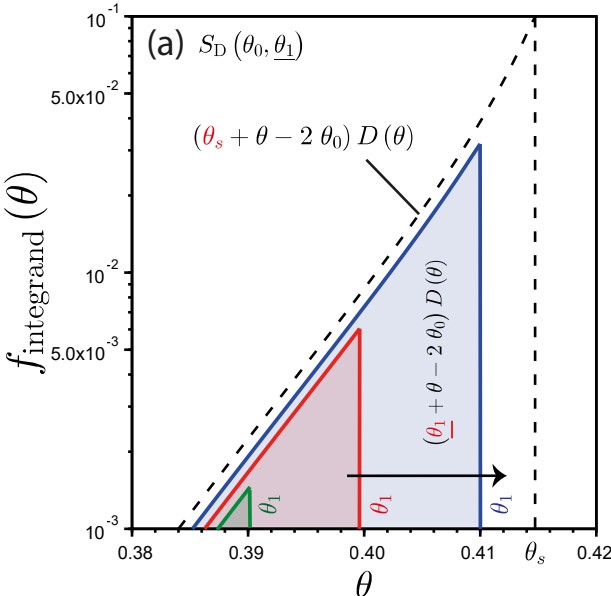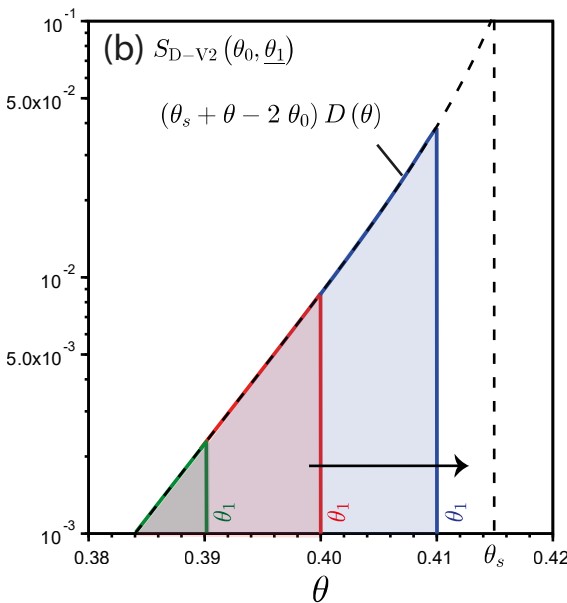

**Figure 3.** Illustration of the regular strategies for the estimation of the limits for the case of saturation $\theta_1 \to \theta_s$, with the estimation of $S_u(\theta_0, \theta_s)$ using either $S_D(\theta_0, \theta_1)$, Eq. 4, (a) or $S_{D-V2}(\theta_0, \theta_1)$, Eq. 22, (b). The integration proceeds from the given value of $\theta_1$ to the left, $\theta \geq \theta_1$, defining the solid zones. The values of $\theta_1$ is increased step by step to reach $S_u(\theta_0, \theta_s)$ (see arrow and the extension of the solid zones). In the equations, the variables that are varied to reach the targeted limits are underlined, and figures are zoomed in in the vicinity of the limits. The differences in equations between the target and the integrated integrands are in red. Illustrative case of the computation of the limits of $S_D(\theta_0, \theta_1)$ or $S_{D-V2}(\theta_0, \theta_1)$ for a synthetic loamy soil with an initial water content of $\theta_0 = 0.25$; successive values are considered for the final water contents, $\theta_1 \in \{0.39, 0.40, 0.41\}$.

As mentioned above for $S_{K-V2}$, $S_{D-V2}$ should be only used for the determination of $S_u(\theta_0, \theta_s)$, and not for the computation of sorptivity corresponding to other values of final water contents, $S_{D-V2}$ integrates the integrand related exclusively to the case of $\theta_1 = \theta_s$.

     The scaled version of these equations can be easily found by applying the scaling equations, Eq. (9-10), to the previous equations, Eqs. (20-21), leading to their scaled versions:

$$S_u^*(S_{e,0}, 1) = \lim_{S_{e,1} \to 1} \sqrt{\int_{S_{e,0}}^{S_{e,1}} \left( \underline{S_{e,1}} + S_e - 2\,S_{e,0} \right) D^*(S_e)\, dS_e}$$

$$= \lim_{S_{e,1} \to 1} S_D^* \left( S_{e,0}, \underline{S_{e,1}} \right) \tag{23}$$

$$S_{\mathrm{u}}^*\left(S_{\mathrm{e},0},1\right) = \lim_{S_{\mathrm{e},1}\to 1} \sqrt{\int_{S_{\mathrm{e},0}}^{S_{\mathrm{e},1}} \left(1 + S_{\mathrm{e}} - 2\,S_{\mathrm{e},0}\right) D^*\left(S_{\mathrm{e}}\right) dS_{\mathrm{e}}}$$

$$= \lim_{S_{\mathrm{e},1}\to 1} S_{\mathrm{D-V2}}^*\left(S_{\mathrm{e},0},\underline{S_{\mathrm{e},1}}\right) \tag{24}$$

with the use of the scaled versions $S_{\mathrm{D}}^*$ and $S_{\mathrm{D-V2}}^*$ of the formulations $S_{\mathrm{D}}$ and $S_{\mathrm{D-V2}}$:

$$\begin{cases} S_{\mathrm{D}}^*\left(S_{\mathrm{e},0},S_{\mathrm{e},1}\right) = \sqrt{\int_{S_{\mathrm{e},0}}^{S_{\mathrm{e},1}} \left(S_{\mathrm{e},1} + S_{\mathrm{e}} - 2\,S_{\mathrm{e},0}\right) D^*\left(S_{\mathrm{e}}\right) dS_{\mathrm{e}}} \\ S_{\mathrm{D-V2}}^*\left(S_{\mathrm{e},0},S_{\mathrm{e},1}\right) = \sqrt{\int_{S_{\mathrm{e},0}}^{S_{\mathrm{e},1}} \left(1 + S_{\mathrm{e}} - 2\,S_{\mathrm{e},0}\right) D^*\left(S_{\mathrm{e}}\right) dS_{\mathrm{e}}} \end{cases} \tag{25}$$

In the following sections, we compare these previously used strategies based on the use of $S_{\mathrm{K}}^*$, and $S_{\mathrm{D}}^*$, and the improved versions designed for the purpose of this study, $S_{\mathrm{K-V2}}^*$ and $S_{\mathrm{D-V2}}^*$ with the proposed mixed formulation $S_{\mathrm{M}}^*$, in terms of accuracy and efficiency.

## 2.3    Validation of estimates against the nominal sorptivity for the selected hydraulic models

### 2.3.1    Hydraulic models and nominal sorptivity

The validation of the computation of sorptivity with the proposed mixed formulation $S_{\mathrm{M}}$ (Eq. 12) and the usual strategies (see Section 2.2) was performed for hydraulic models that present challenging features. Besides, these models are commonly used for the hydraulic characterization of soils:

- The Brooks and Corey (BC) model (Brooks and Corey, 1964) was among the first hydraulic models of soil physics (Hillel, 1998). It uses power law relationships to define the water retention (WR) and hydraulic conductivity (HC) functions and is often considered for integrating sorptivity and finding analytical solutions for water infiltration into soils (e.g., Varado et al., 2006). The **BC model** reads as follows:

$$\begin{cases} \theta_{\mathrm{BC}}(h) = \begin{cases} \theta_{\mathrm{s}} & h \geq h_{\mathrm{BC}} \\ \theta_{\mathrm{r}} + (\theta_{\mathrm{s}} - \theta_{\mathrm{r}}) \left(\frac{h_{\mathrm{BC}}}{h}\right)^{\lambda_{\mathrm{BC}}} & h < h_{\mathrm{BC}} \end{cases} \\ K_{\mathrm{BC}}(\theta) = K_{\mathrm{s}} \left(\frac{\theta - \theta_{\mathrm{r}}}{\theta_{\mathrm{s}} - \theta_{\mathrm{r}}}\right)^{\eta_{\mathrm{BC}}} \end{cases} \tag{26}$$

- The van Genuchten – Burdine (vGB) model combines van Genuchten (1980) model with Burdine condition ($m = 1 - \frac{2}{n}$) for the WR function and the Brooks and Corey (1964) model for the HC function. It was the basis of the development of BEST methods and often considered for the hydraulic characterization of soils (Lassabatere et al., 2006; Yilmaz et al., 2010; Bagarello et al., 2014a). These formulations are considered to be one of the most consistent to use for modeling

water infiltration into soils (Fuentes et al., 1992). The **vGB model** reads as follows:

$$\begin{cases} \theta_{\mathrm{vGB}}(h) = \theta_{\mathrm{r}} + (\theta_{\mathrm{s}} - \theta_{\mathrm{r}}) \left(1 + \left(\frac{h}{h_{\mathrm{vGB}}}\right)^{n_{\mathrm{vGB}}}\right)^{-m_{\mathrm{vGB}}} \\ m_{\mathrm{vGB}} = 1 - \frac{2}{n_{\mathrm{vGB}}} \\ K_{\mathrm{vGB}}(\theta) = K_{\mathrm{s}} \left(\frac{\theta - \theta_{\mathrm{r}}}{\theta_{\mathrm{s}} - \theta_{\mathrm{r}}}\right)^{\eta_{\mathrm{vGB}}} \end{cases} \tag{27}$$

– The van Genuchten – Mualem (vGM) model combines van Genuchten (1980) model with Mualem's condition ($m = 1 - \frac{1}{n}$) for the WR function and the Mualem (1976) capillary model for the HC function. The vGM model is among the most widely-used models, in particular for the numerical modeling of flow in the vadose zone (Šimůnek et al., 2003). The **vGM model** reads as follows:

$$\begin{cases} \theta_{\mathrm{vGM}}(h) = \theta_{\mathrm{r}} + (\theta_{\mathrm{s}} - \theta_{\mathrm{r}}) \left(1 + \left(\frac{h}{h_{\mathrm{vGM}}}\right)^{n_{\mathrm{vGM}}}\right)^{-m_{\mathrm{vGM}}} \\ m_{\mathrm{vGM}} = 1 - \frac{1}{n_{\mathrm{vGM}}} \\ K_{\mathrm{vGM}}(\theta) = K_{\mathrm{s}} \left(\frac{\theta - \theta_{\mathrm{r}}}{\theta_{\mathrm{s}} - \theta_{\mathrm{r}}}\right)^{l_{\mathrm{vGM}}} \left(1 - \left(1 - \left(\frac{\theta - \theta_{\mathrm{r}}}{\theta_{\mathrm{s}} - \theta_{\mathrm{r}}}\right)^{\frac{1}{m_{\mathrm{vGM}}}}\right)^{m_{\mathrm{vGM}}}\right)^2 \end{cases} \tag{28}$$

– The Kosugi (KG) model relates the WR function to the soil pore size distribution assuming log-normal distributions (Kosugi, 1996). It is quite popular as the consequence of its physical meaning and soundness (Pollacco et al., 2013; Nasta et al., 2013) and was also recently implemented into BEST methods for the hydraulic characterization of soils (Fernández-Gálvez et al., 2019). The **KG model** reads as follows:

$$\begin{cases} \theta_{\mathrm{KG}}(h) = \theta_{\mathrm{r}} + \frac{(\theta_{\mathrm{s}} - \theta_{\mathrm{r}})}{2} \mathrm{erfc}\left(\frac{\ln\left(\frac{h}{h_{\mathrm{KG}}}\right)}{\sqrt{2}\sigma_{\mathrm{KG}}}\right) \\ \\ K_{\mathrm{KG}}(\theta) = K_{\mathrm{s}} \left(\frac{\theta - \theta_{\mathrm{r}}}{\theta_{\mathrm{s}} - \theta_{\mathrm{r}}}\right)^{l_{\mathrm{KG}}} \left(\frac{1}{2} \mathrm{erfc}\left(\mathrm{erfc}^{-1}\left(2\frac{\theta - \theta_{\mathrm{r}}}{\theta_{\mathrm{s}} - \theta_{\mathrm{r}}}\right) + \frac{\sigma_{\mathrm{KG}}}{\sqrt{2}}\right)\right)^2, \end{cases} \tag{29}$$

where erfc stands for the complementary error function.

These models involve the following common scale hydraulic parameters: residual water content, $\theta_{\mathrm{r}}$, saturated water content, $\theta_{\mathrm{s}}$, scale parameter for the water pressure head, $h_{\mathrm{g}}$, ($h_{\mathrm{BC}}$, $h_{\mathrm{vGB}}$, $h_{\mathrm{vGM}}$, or $h_{\mathrm{KG}}$), and saturated hydraulic conductivity, $K_s$. The BC models involve a non-null air-entry water pressure head, $h_{\mathrm{BC}}$, meaning that air needs a given suction to enter into the soil and to desaturate the soil. For the sake of simplicity, the scale parameter for water pressure head is often fixed at the air-entry pressure head, so that $h_{\mathrm{g}} = h_{\mathrm{BC}}$. In addition, these hydraulic models involve one or two shape parameters for each set of WR and HC functions, specifically $\lambda_{\mathrm{BC}}$ and $\eta_{\mathrm{BC}}$ for the BC model, $m_{\mathrm{vGB}}$, $n_{\mathrm{vGB}}$ and $\eta_{\mathrm{vGB}}$ for the vGB model, $m_{\mathrm{vGM}}$, $n_{\mathrm{vGM}}$ and $l_{\mathrm{vGM}}$ for the vGM model, and, lastly, $\sigma_{\mathrm{KG}}$ and $l_{\mathrm{KG}}$ for the KG model. In order to simplify these equations and to reduce the risk of equifinality and non-unique optimization when inverting (Pollacco et al., 2013), the following capillary model has been proposed to link these shape parameters (Haverkamp et al., 2005):

$$\eta = \frac{2}{\lambda} + 2 + p \tag{30}$$

where $\lambda = \lambda_{\mathrm{BC}}$ for the BC model and $\lambda = mn$ for the vGB models. Besides, the shape parameters $l_{\mathrm{vGM}}$ and $l_{\mathrm{KG}}$ are usually fixed at $\frac{1}{2}$. In this study, the computations are performed considering the relationship given by Eq. (30).

The application of the scaling procedure Eq. (9) to these hydraulic models, i.e., Eqs. (26)-(29) leads to the following scaled hydraulic models (Lassabatere et al., 2021):

BC model:


$$
\begin{cases}
S_{e,\mathrm{BC}}(h^*) = (1 - H(1+h^*))|h^*|^{-\lambda_{\mathrm{BC}}} + H(1+h^*) \\
K_{r,\mathrm{BC}}(S_e) = S_e^{\eta_{\mathrm{BC}}}
\end{cases}
\tag{31}
$$

vGB model:

$$
\begin{cases}
S_{e,\mathrm{vGB}}(h^*) = (1 + |h^*|^{n_{\mathrm{vGB}}})^{-m_{\mathrm{vGB}}} \\
\text{with } m_{\mathrm{vGB}} = 1 - \dfrac{2}{n_{\mathrm{vGB}}} \\
K_{r,\mathrm{vGB}}(S_e) = S_e^{\eta_{\mathrm{vGB}}}
\end{cases}
\tag{32}
$$

vGM model:

$$
\begin{cases}
S_{e,\mathrm{vGM}}(h^*) = (1 + |h^*|^{n_{\mathrm{vGM}}})^{-m_{\mathrm{vGM}}} \\
\text{with } m_{\mathrm{vGM}} = 1 - \dfrac{1}{n_{\mathrm{vGM}}} \\
K_{r,\mathrm{vGM}}(S_e) = S_e^{l_{\mathrm{vGM}}}\left(1 - \left(1 - S_e^{\frac{1}{m_{\mathrm{vGM}}}}\right)^{m_{\mathrm{vGM}}}\right)^2
\end{cases}
\tag{33}
$$

KG model:

$$
\begin{cases}
S_{e,\mathrm{KG}}(h^*) = \frac{1}{2}\mathrm{erfc}\left(\dfrac{\ln(|h^*|)}{\sqrt{2}\sigma_{\mathrm{KG}}}\right) \\
K_{r,\mathrm{KG}}(S_e) = S_e^{l_{\mathrm{KG}}}\left(\frac{1}{2}\mathrm{erfc}\left(\mathrm{erfc}^{-1}(2S_e) + \dfrac{\sigma_{\mathrm{KG}}}{\sqrt{2}}\right)\right)^2.
\end{cases}
\tag{34}
$$

These hydraulic models have the following hydraulic diffusivity functions, $D^*(S_e) = K(S_e)\frac{dh^*}{dS_e}$ (Lassabatere et al., 2021):

$$
\begin{cases}
D^*_{\mathrm{BC}}(S_e) = \dfrac{1}{\lambda_{\mathrm{BC}}} S_e^{\eta_{\mathrm{BC}} - \left(\frac{1}{\lambda_{\mathrm{BC}}}+1\right)} \\
D^*_{\mathrm{vGB}}(S_e) = \dfrac{1-m_{\mathrm{vGB}}}{2\,m_{\mathrm{vGB}}} S_e^{\eta_{\mathrm{vGB}} - \frac{1+m_{\mathrm{vGB}}}{2\,m_{\mathrm{vGB}}}}\left(1 - S_e^{\frac{1}{m_{\mathrm{vGB}}}}\right)^{-\frac{1+m_{\mathrm{vGB}}}{2}}. \\
D^*_{\mathrm{vGM}}(S_e) = \dfrac{1-m_{\mathrm{vGM}}}{m_{\mathrm{vGM}}} S_e^{l_{\mathrm{vGM}} - \frac{1}{m_{\mathrm{vGM}}}}\left(\left(1 - S_e^{\frac{1}{m_{\mathrm{vGM}}}}\right)^{-m_{\mathrm{vGM}}} + \left(1 - S_e^{\frac{1}{m_{\mathrm{vGM}}}}\right)^{m_{\mathrm{vGM}}} - 2\right) \\
D^*_{\mathrm{KG}}(S_e) = \frac{1}{2}\sqrt{\frac{\pi}{2}}\sigma_{\mathrm{KG}} S_e^{l_{\mathrm{KG}}}\left(erfc\left(erfc^{-1}(2S_e) + \frac{\sigma_{\mathrm{KG}}}{\sqrt{2}}\right)\right)^2 e^{\left(erfc^{-1}(2S_e)\right)^2 + \sqrt{2}\sigma_{\mathrm{KG}}\,erfc^{-1}(2S_e)}
\end{cases}
\tag{35}
$$

These equations are needed for the computation of the dimensionless sorptivity with the proposed mixed formulation $S_{\mathrm{M}}^{*2}$, and
the regular formulations $S_{\mathrm{K}}^{*2}$, $S_{\mathrm{K-V2}}^{*2}$, $S_{\mathrm{D}}^{*2}$ and $S_{\mathrm{D-V2}}^{*2}$.

The studied hydraulic models exhibit contrasting and challenging features for the computation of sorptivity, including non-null water pressure heads $h_a^* < 0$, and infinite hydraulic diffusivity close to saturation $\lim_{S_e \to 1} D^*(S_e) = +\infty$. The complexity may also increase with the values of related shape parameters. Therefore, Lassabatere et al. (2021) define a shape index to characterize the spread of the WR functions around $S_e(h^*) = \frac{1}{2}$. Regardless of the chosen hydraulic model, the values of
$x$ close to zero correspond to a large spread of WR functions with a smooth descent of the saturation degree, $S_e$, with the increase of $|h^*|$ (See Fig. 2, in Lassabatere et al. (2021), and also section 3). Conversely, when $x$ gets close to unity, WR

functions approach the stepwise function with an abrupt decrease of $S_e$ with the increase of $|h^*|$. Lassabatere et al. (2021) defined the WR shape index $x$ as follows:

$$\begin{cases} x_{\mathrm{BC}} = \frac{\lambda_{\mathrm{BC}}}{2+\lambda_{\mathrm{BC}}} \\[2mm] x_{\mathrm{vGB}} = m_{\mathrm{vGB}} \\[2mm] x_{\mathrm{vGM}} = m_{\mathrm{vGM}} \\[2mm] x_{\mathrm{KG}} = \frac{1}{1+\sigma_{\mathrm{KG}}} \end{cases} \tag{36}$$

Lassabatere et al. (2021) also analytically determined the maximum squared scaled sorptivity $S^{*2}(-\infty, 0)$, also referred to as the parameter $c_{\mathrm{p}}$, as a function of the WR shape index $x$, for the BC, vBG, and vGM models:

$$\begin{cases} c_{\mathrm{p,BC}}(x) = 2 + \frac{1-x}{5x+1} + \frac{1-x}{7x+1} \\[2mm] c_{\mathrm{p,vGB}}(x) = \Gamma\left(\frac{3-x}{2}\right)\left[\frac{\Gamma\left(\frac{1+5x}{2}\right)}{\Gamma(1+2x)} + \frac{\Gamma\left(\frac{1+7x}{2}\right)}{\Gamma(1+3x)}\right] \\[2mm] c_{\mathrm{p,vGM}}(x) = \Gamma(2-x)\left[\frac{\Gamma\left(\frac{3}{2}x\right)}{\left(\frac{3}{2}x-1\right)\Gamma\left(\frac{1}{2}x\right)} + \frac{\Gamma\left(\frac{5}{2}x\right)}{\left(\frac{5}{2}x-1\right)\Gamma\left(\frac{3}{2}x\right)}\right] + (x-1)\left[\frac{\Gamma\left(\frac{3}{2}x\right)\Gamma(1+x)}{\left(\frac{3}{2}x-1\right)\Gamma\left(\frac{5}{2}x\right)} + \frac{\Gamma\left(\frac{5}{2}x\right)\Gamma(1+x)}{\left(\frac{5}{2}x-1\right)\Gamma\left(\frac{7}{2}x\right)} - 2\left(\frac{1}{\frac{3}{2}x-1} + \frac{1}{\frac{5}{2}x-1}\right)\right] \end{cases} \tag{37}$$

Note that no analytical formulation can be found for the case of Kosugi's hydraulic model, so $c_{\mathrm{p,KG}}(x)$ must be computed numerically (Lassabatere et al., 2021). Note also that in Eqs. (37), nominal sorptivities are defined with the use of the capillary

model Eq. (30) for the BC and vGB models, and $l_{\mathrm{vGM}} = l_{\mathrm{KG}} = \frac{1}{2}$.

### 2.3.2   Paper methodology and computations

In this study, we argue the following points: (i) the studied hydraulic models for WR and HC functions exhibit challenging conditions for the computation of sorptivity, (ii) the proposed mixed formulation is an ideal estimator for sorptivity, (iii) the usual methods, based on the use of $S_{\mathrm{K}}$ and $S_{\mathrm{D}}$ (Eq. 5 and Eq. 4), or their improved version, $S_{\mathrm{K-V2}}$ and $S_{\mathrm{D-V2}}$ (Eq. 16

and Eq. 22) do not necessarily provide accurate estimations of the targeted nominal sorptivity, and (iv) errors in sorptivity estimation may drastically impact its use for modeling water infiltration into soils. To demonstrate these points, we consider the following conditions. Firstly, we only investigate the case of the scaled sorptivity. Indeed, if we let be any estimator $\hat{S}$ of the nominal dimensional sorptivity $S$, and the related scaled variables, $\hat{S}^*$ and $S^*$, the following relations emerge:

$$\begin{aligned} E_{\mathrm{r}}(S) &= \frac{\hat{S}-S}{S} \\[2mm] &= \frac{\hat{S}^* \cdot \sqrt{h_{\mathrm{g}}K_{\mathrm{s}}(\theta_{\mathrm{s}}-\theta_{\mathrm{r}})} - S^* \cdot \sqrt{h_{\mathrm{g}}K_{\mathrm{s}}(\theta_{\mathrm{s}}-\theta_{\mathrm{r}})}}{S^* \cdot \sqrt{h_{\mathrm{g}}K_{\mathrm{s}}(\theta_{\mathrm{s}}-\theta_{\mathrm{r}})}} \\[2mm] &= \frac{\hat{S}^* - S^*}{S^*} \\[2mm] &= E_{\mathrm{r}}(S^*), \end{aligned} \tag{38}$$

proving that the accuracy of the scaled estimator corresponds exactly to the accuracy of the dimensional estimator. Next, we consider the maximum scaled sorptivity, $S^* (-\infty, 0)$, since it involves at the same time the two types of challenges, i.e., very dry initial conditions with infinite water pressure head, $h_0 = -\infty$, and saturated final conditions with null water pressure head, $h_1 = 0$. Then, the modeling of water infiltration into soils is illustrated for a loamy soil and relies on the use of dimensional sorptivity (Section 3.4).

The first step, point (i), involves the study of the selected models with regard to the shapes of WR and HC functions. We computed the WR and HC functions for the four selected models, considering the following values of the WR shape index: $x \in \{0.01, 0.02, ..., 0.99\}$. For the second goal of the study, point (ii), we compared the values provided by the proposed mixed formulation $S_{\mathrm{M}}^*$ with the nominal (error-free) values of sorptivity, $S^* = S^* (-\infty, 0) = \sqrt{c_{\mathrm{p}}}$, provided by the exact analytical formulations (Eqs. 37) for BC, vGB and vGM models. The computations were performed for all the values of the WR shape

index $x$, and the accuracy of $S_{\mathrm{M}}^*$ was discussed as a function of $x$.

For the third goal, point (iii), the estimations provided by the usual strategies were compared to the estimates provided by the proposed mixed formulation, $S_{\mathrm{M}}^*$, to evaluate the efficiency of those previously used strategies. We considered several scenarios for the use of $S_{\mathrm{K}}^*$, $S_{\mathrm{K-V2}}^*$, $S_{\mathrm{D}}^*$, $S_{\mathrm{D-V2}}^*$, with several values of the lower water pressure head $h_0^*$ used in Eq. (17) and Eq. (18), and several values of the final saturation degree $S_{e,1}$ used in Eq. (23) and Eq. (24). We also investigated the estimation

error as a function of the shape parameters for the four studied hydraulic models (BC, vGB, vGM, and KG).

Finally, regarding point (iv), we computed the dimensional sorptivity for a synthetic loamy soil, $S(h_0, h_1)$, using both the mixed formulation and the regular methods, to investigate its dependency on both initial and final conditions and hydraulic models (BC, vGM and KG). For the sake of clarity, we omitted the case of the vGB model and considered the three more contrasting models. Then, we investigated its use for the modeling of water infiltration into loamy soil. For that last purpose, we

applied the different calculated sorptivity values to analytical models for the modeling of cumulative infiltrations corresponding to a set of water pressure head imposed at the soil surface ($h_{\mathrm{f}} = -150\,\mathrm{mm}$ or $h_{\mathrm{f}} = 30\,\mathrm{mm}$) along with an initial water pressure head of -10 m ($h_{\mathrm{i}} = -10^4\,\mathrm{mm}$). The analytical models used to model the cumulative infiltrations are described in the Results section. The BC, vGM, and KG models were fitted to the hydraulic functions of the loamy soil, as defined by Carsel and Parrish (1988), before computation of sorptivity to investigate the impact of the choice of the hydraulic models on sorptivity

estimation and related impact on cumulative infiltrations. The different synthetic cumulative infiltrations were discussed with regard to the sorptivity estimation method, the choice of hydraulic models for the WR and HC functions, and the choice of the analytical model. All the computations were performed using Scilab software (Campbell et al., 2010) and are available on the Zenodo website (Lassabatere, 2022).

## 3   Results

### 3.1   Analysis of the selected hydraulic models and related challenging features

This section presents the four sets of models for describing the water retention and hydraulic conductivity functions, their features and their dependency on related shape parameters. The features of the WR-HC functions are determinant with regards

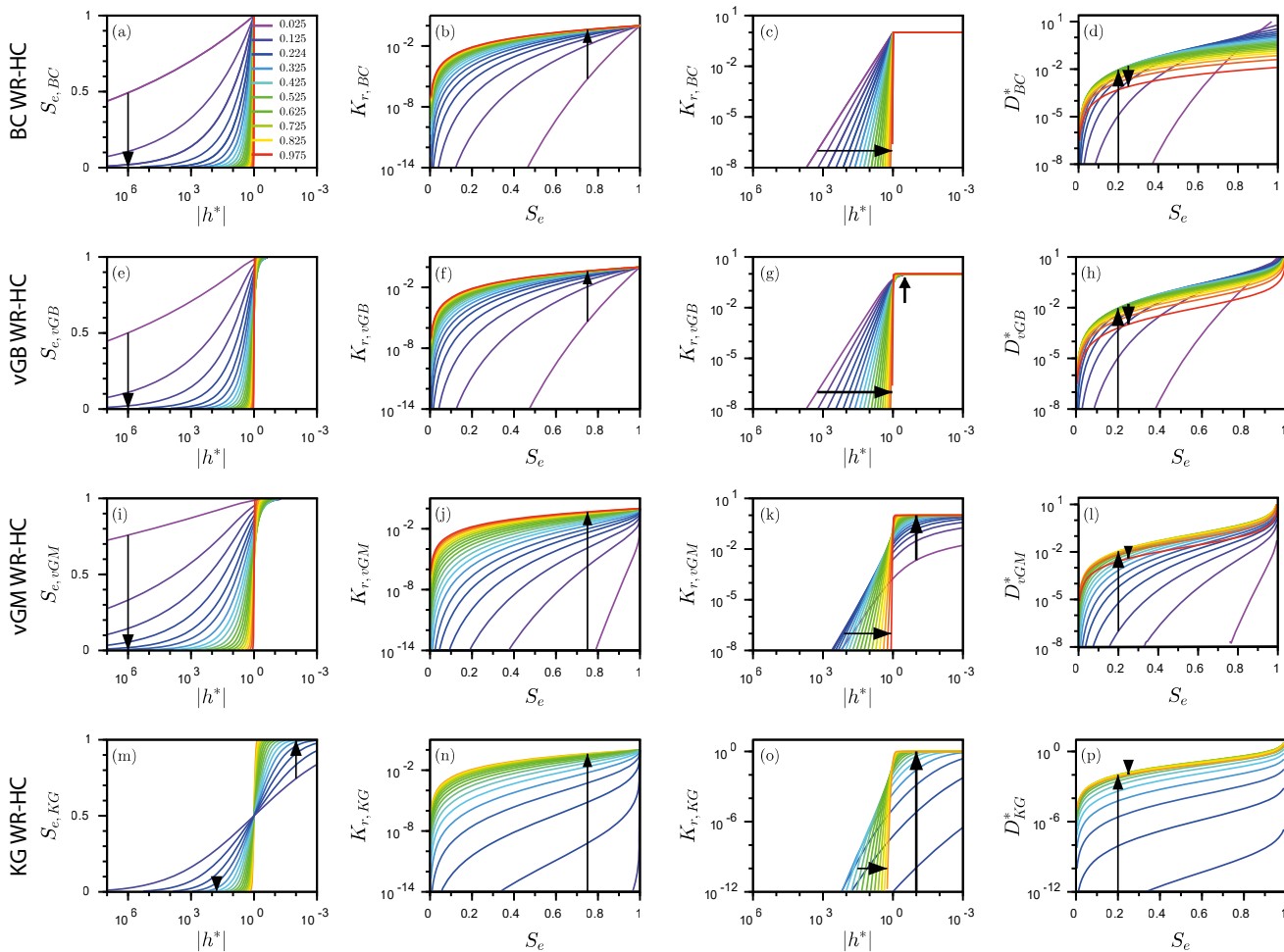

**Figure 4.** Water retention (WR) and hydraulic conductivity (HC) curves for different values of the WR shape index $x$. The first column shows WR as $S_e(h^*)$, the second column shows HC as $K_r(S_e)$, the third column shows HC as $K_r(h^*)$, and the fourth column shows diffusivity as $D^*(S_e)$; the four tested models include Brooks and Corey (BC) ($1^{st}$ row), van Genuchten – Burdine (vGB) ($2^{nd}$ row), van Genuchten-Mualem (vGM) ($3^{rd}$ row), and Kosugi (KG) models ($4^{rd}$ row). The arrows indicate the trends with increasing WR shape index $x$. The hydraulic parameters $\lambda_{\text{BC}}$, $m_{\text{vGM}}$, $m_{\text{vGB}}$, and $\sigma_{\text{KG}}$ were computed as a function of $x$ using Eq. (36) with $l_{\text{vGM}} = l_{\text{KG}} = \frac{1}{2}$. Adapted from Lassabatere et al. (2021).

to the estimation of sorptivity. The BC model has a non-null air-entry water pressure head (Fig. 4a, shown by the plateau for $|h^*| \leq 1$, i.e., $h^* \geq -1$), meaning that the sorptivity has a non-null saturated part that must be accounted for. This condition is one of the challenging features of the diffusivity form of sorptivity, Eq. (4). Conversely, the three other hydraulic models do not have any air-entry water pressure head values (Fig. 4e, i, m do not have plateaus), but rather have infinite values of the hydraulic diffusivity close to saturation, thus posing potential problems of convergence (Fig. 4h, l, p). The use of these

models allows us to characterize the improvements offered by $S_M^*$ compared to the usual use of $S_D^*$ for the cases of problematic computation close to saturation (i.e., $h_1 \to 0$ and $\theta_1 \to \theta_s$). Similarly, accuracy of the commonly used $S_K^*$ version can be challenged when integrating over infinite intervals $(-\infty, 0]$, particularly for hydraulic models that are characterized by a slow decrease in the saturation degree, $S_e$, for quasi-infinite water pressure heads, $h^*$. The chosen hydraulic models are thus expected to be challenging, in particular the BC, vGB, and vGM models, which keep high values of saturation degrees even for quasi-infinite water pressure heads (Fig. 4a, e, i). The KG model, which is more symmetrical and characterized by a larger decrease in $S_e$ when $h^*$ decreases (Fig. 4m) is expected to be less challenging. Regarding those challenges, the shapes of the WR, HC, and hydraulic diffusivity functions are also of importance. We expect the small values of $x$ to be more problematic, in particular with the use of $S_K^*$, due to the smooth decrease in $S_e$ with $h^*$ (Fig. 4, first column). Conversely, we expect more problems with the use of $S_D^*$ for large values of $x$, with quasi infinite values for the hydraulic diffusivity close to saturation, i.e., $S_e \to 1$ (Fig. 4h, l, p).

### 3.2 Validation of the proposed mixed formulation, $S_M^*$, against the nominal sorptivity, $S^*(-\infty, 0)$

The computations using $S_M^*$ (Eq. 12-13) with $h_c^* = h^* \left( \frac{S_{e,0} + S_{e,1}}{2} \right)$ were efficient in most cases, regardless of the value of the WR shape index $x$. The use of the threshold $10^z$ was necessary for the first value of the WR shape index $x$ for the BC model ($x_{BC} = 0.01$), the first two values for the vGB model ($x_{vGB} \in \{0.01, 0.02\}$) and the first 17 values for the vGM model ($x_{vGM} \in \{0.01, 0.02, .., 0.17\}$). The value of $z = 0$ was enough to allow the computation in all cases, apart from the case of vGM model for which the value of $z = -1$ had to be considered for $x_{vGM} \in \{0.02, .., 0.07\}$ and $z = -2$ for $x_{vGM} = 0.01$. Note that, as long as convergence is obtained, the values of $S_M^*(-\infty, 0)$ do not depend on $z$. With this strategy involving a threshold, the proposed mixed formulation, $S_M^*$, provides estimates for all cases, i.e., for all hydraulic models and all values of the WR shape index, $x$. A sensitivity analysis was also performed for the KG model and led to the same success with $z = 0$ regardless of the value of the WR shape index $x_{KG}$ (data not shown).

Relative error, $E_r$, between the estimates provided by the proposed mixed formulation, $S_M^*(-\infty, 0)$, and the targeted sorptivity, $S^*(-\infty, 0) = \sqrt{c_p}$, were analyzed in terms of means, standard deviations, and minimum and maximum values (Table 1):

$$E_r = \frac{S_M^*(-\infty, 0) - \sqrt{c_p}}{\sqrt{c_p}} \tag{39}$$

The accuracy of the proposed mixed formulation, $S_M^*$, Eqs. (12-13), is excellent for all the models and all values of WR shape index $x$ (Table 1). The average relative errors are on the order of $10^{-13}$ for the BC and vGB models, and on the order of $10^{-9}$ for the vGM model (Table 1, $\overline{E_r}$). The minimum errors were $< 10^{-15}$ for all the models (Table 1, min). The maximum errors were $\approx 10^{-12}$ for the BC and the vGB models and $\approx 10^{-7}$ for the vGM model (Table 1, max). In other words, the mixed formulation, $S_M^*$, provides extremely accurate estimations of the targeted scaled sorptivity $S^*(-\infty, 0)$. The proposed mixed formulation, $S_M^*$, can therefore be considered to be an excellent estimator of sorptivity in all cases, regardless the choice of the hydraulic model and related values of shape parameters.

**Table 1.** Absolute values of relative errors, $|E_r|$, between the proposed mixed formulation, $S_M^*$, and the targeted scaled sorptivity, $S^*(-\infty, 0) = \sqrt{c_p}$, with the mean $\overline{|E_r|}$, the standard deviation ($\sigma_{|E_r|}$), the minimum and the maximum values for the three hydraulic models whose sorptivity in analytically tractable: Brooks and Corey (BC), van Genuchten – Burdine (vGB), van Genuchten-Mualem (vGM). Note that $10^{-16}$ corresponds to the relative precision of numbers in scilab.

| $|E_r|$ | BC | vGB | vGM |
|---|---|---|---|
| $\overline{|E_r|}$ | $1.445\,10^{-13}$ | $5.594\,10^{-13}$ | $3.309\,10^{-9}$ |
| $\sigma_{|E_r|}$ | $2.774\,10^{-13}$ | $1.014\,10^{-12}$ | $1.934\,10^{-8}$ |
| min | $< 10^{-16}$ | $< 10^{-16}$ | $8.255\,10^{-16}$ |
| max | $1.201\,10^{-12}$ | $6.037\,10^{-12}$ | $2.000\,10^{-7}$ |

## 3.3 Study of the usual strategies for estimating the nominal sorptivity, $S^*(-\infty, 0)$

In this section, we compare the estimates provided by the strategies commonly considered (i.e., Eqs. (4-5), $S_D^*$, $S_{D-V2}^*$, $S_K^*$, and $S_{K-V2}^*$) with the reference values of the targeted sorptivity $S^* = S^*(-\infty, 0)$. For these comparisons, we consider the analytical formulations of Eq. (37) for the BC, vGB, and vGM models and use the proposed mixed formulation $S_M^*$ for the KG model. Indeed, no analytical expressions are available for this last model, whereas the estimates provided by $S^* = S_M^*$ are very convincing for the BC, vGB, and vGM models (see Section 3.2) and thus $S_M^*$ is assumed to be as accurate for the KG model.

### 3.3.1 Illustrative example

In this example, we investigate the accuracy of the limits of the functions $S_K^*$ and $S_{K-V2}^*$ towards $S^*$. These functions, in particular $S_K^*$, are often used without special attention regarding their accuracy. $S_K^*$ and $S_{K-V2}^*$ converge to the limit $S^*$ when $|h_0^*|$ becomes large enough (Fig. 5, left column). For instance, for the BC model with $\lambda = 0.56$, the use of $S_K^*$ with $h_0^* = -10$ and $h_0^* = -100$ has respective relative errors of $E_r = -4.1\%$ and $E_r = -15\%$ (Figure 5a, red dashed line). The second estimator, $S_{K-V2}^*$, converges much faster than $S_K^*$. With the same values of $h_0^*$, the relative errors drop below $0.01\%$ in absolute value (Figure 5a, blue continuous line). In this case, the convergence towards the target can be achieved by integrating from $h_0^* = -10$, with high accuracy. Such an improvement results from setting the initial saturation degree, $S_{e,0}$, at its target value, $S_{e,0} = 0$, in the integrand (see Eq. 18 versus Eq. 17). The same conclusions can be stated regardless of the selected hydraulic models (Figure 5c, e, g).

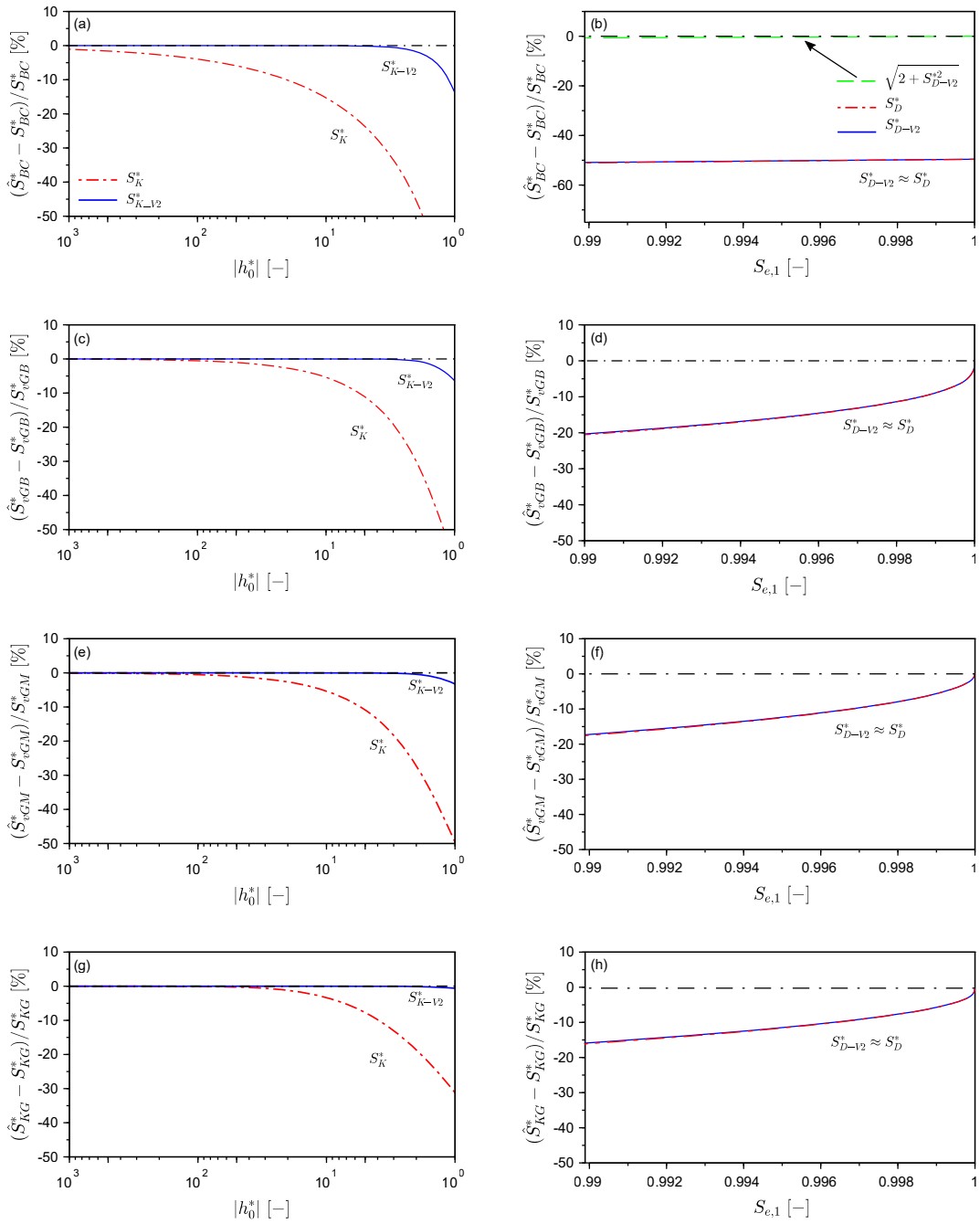

**Figure 5.** : Relative errors of the regular functions $S_D^*(0, S_{e,1})$ and $S_{D-V2}^*(0, S_{e,1})$ (right) and $S_K^*(h_0^*, 0)$ and $S_{K-V2}^*(h_0^*, 0)$ (left) towards $S^*(-\infty, 0)$ for the four hydraulic models, for the following specific cases: BC model with $\lambda_{BC} = 56$ ($x_{BC} = 0.22$), vGB model with $n_{vGB} = 3$ ($x_{vGB} = 1/3$), vGM model with $n_{vGM} = 2$ ($x_{vGM} = 1/2$) with $l_{vGM} = 0.5$, and KG model with $\sigma_{KG} = 1.5$ ($x_{KG} = 0.4$) and $l_{KG} = 0.5$.

The convergence of the two functions, $S_\mathrm{D}^*$ and $S_{\mathrm{D-V2}}^*$, are depicted for the case of BC functions in Figure 5b. For both estimators, the estimates are far from the target values, $S^*$, with $|E_\mathrm{r}|$ close to $50\%$. This large error results from the omission of the saturated part of sorptivity, as explained above (Section 2.2). By design, $S_\mathrm{D}^*$ and $S_{\mathrm{D-V2}}^*$ converge towards the unsaturated part of sorptivity, $S_u\left(0,1\right) = \sqrt{\int_0^1 \left(1 + S_\mathrm{e}\right) D^*\left(S_\mathrm{e}\right) dS_\mathrm{e}}$, and miss the additional saturated part of the scaled sorptivity, $2\left(S_{e,1} - S_{e,0}\right) K_r\left(S_{e,1}\right)\left(h_1^* - h_\mathrm{a}^*\right) = 2$, given that $h_1^* = 0$ $(S_{e,1} = 1)$, $h_0^* = -\infty$ $(S_{e,0} = 0)$, and $h_\mathrm{a}^* = 1$. When the saturated portion is added to the estimators, the convergence becomes excellent (Figure 5b, green line). These results show that the computation of sorptivity using the integration with regards to saturation degree, like Eq. (4) and Eq. (25), leads to erroneous estimations. Accounting for the term $2(\theta_s - \theta_0) K_s |h_a|$ in Eq. (6) is thus essential. Note that Eq. (4) is often considered and used in most studies, even though it can lead to large under-estimation of sorptivity for the case of non-null air-entry water pressure heads. That factor is of prime importance regarding the accurate estimation of sorptivity.

For the other hydraulic models, the estimators $S_\mathrm{D}^*$ and $S_{\mathrm{D-V2}}^*$ are significantly better (Figure 5d, f, h). Indeed, the air-entry water pressure head is null in these models, which removes the large underestimation due to the omission of the saturated part of sorptivity, as observed for BC model. However, the under-estimation remains substantial, with relative errors on the order of $15 - 20\%$ for $S_{e,1} = 0.99$ (Figure 5d, f, h). Even when $S_{e,1} = 0.999$, the relative errors are larger than $10\%$. From these results, we conclude that the determination of the sorptivity, $S^*$, by approaching the final saturation degree to unity, does not provide good estimates, regardless of the estimators considered. These poor estimates result from the fact that the diffusivity, and thus the integrand, are infinite. The convergence of the integrals defined by Eqs. (25) and thus Eq. (4) are slow and prevent accurate estimations. Conversely to the case of $S_\mathrm{K}^*$ and $S_{\mathrm{K-V2}}^*$, changing the integrand by fixing it to the targeted integrand does not significantly improve the convergence.

### 3.3.2  Relative errors as a function of the WR shape index $x$

The previous results were presented for specific values of the shape parameters in Figure 5. One may wonder if the accuracy of estimators $S_\mathrm{K}^*$ and $S_{\mathrm{K-V2}}^*$, on the one hand, and $S_\mathrm{D}^*$ and $S_{\mathrm{D-V2}}^*$, on the other, varies with the shape parameters and indices. Figure 6 depicts the relative errors of the estimators $S_\mathrm{K}^*$ and $S_{\mathrm{K-V2}}^*$ as a function of the shape parameters for different values of $h_0^*$ (left column of Figure 6) and of the estimators $S_\mathrm{D}^*$ and $S_{\mathrm{D-V2}}^*$ for different values of $S_{e,1}$ (right column of Figure 6). For all hydraulic models except the BC model, the best estimates for $S_\mathrm{D}^*$ and $S_{\mathrm{D-V2}}^*$ are obtained for intermediate values of WR shape indexes (Figure 6, right column). Discrepancies increase for both small or large values of the WR shape index. As detailed above (section 3.3.1), accurate predictions require the practitioner to fix the upper integration boundary, $S_{e,1}$, to 0.999 at least, to get errors less than $10\%$. However, estimates remain poor, even with $S_{e,1} = 0.999$ or $S_{e,1} = 0.9999$, when the WR shape indices tend towards unity (Figure 6d, f, h).

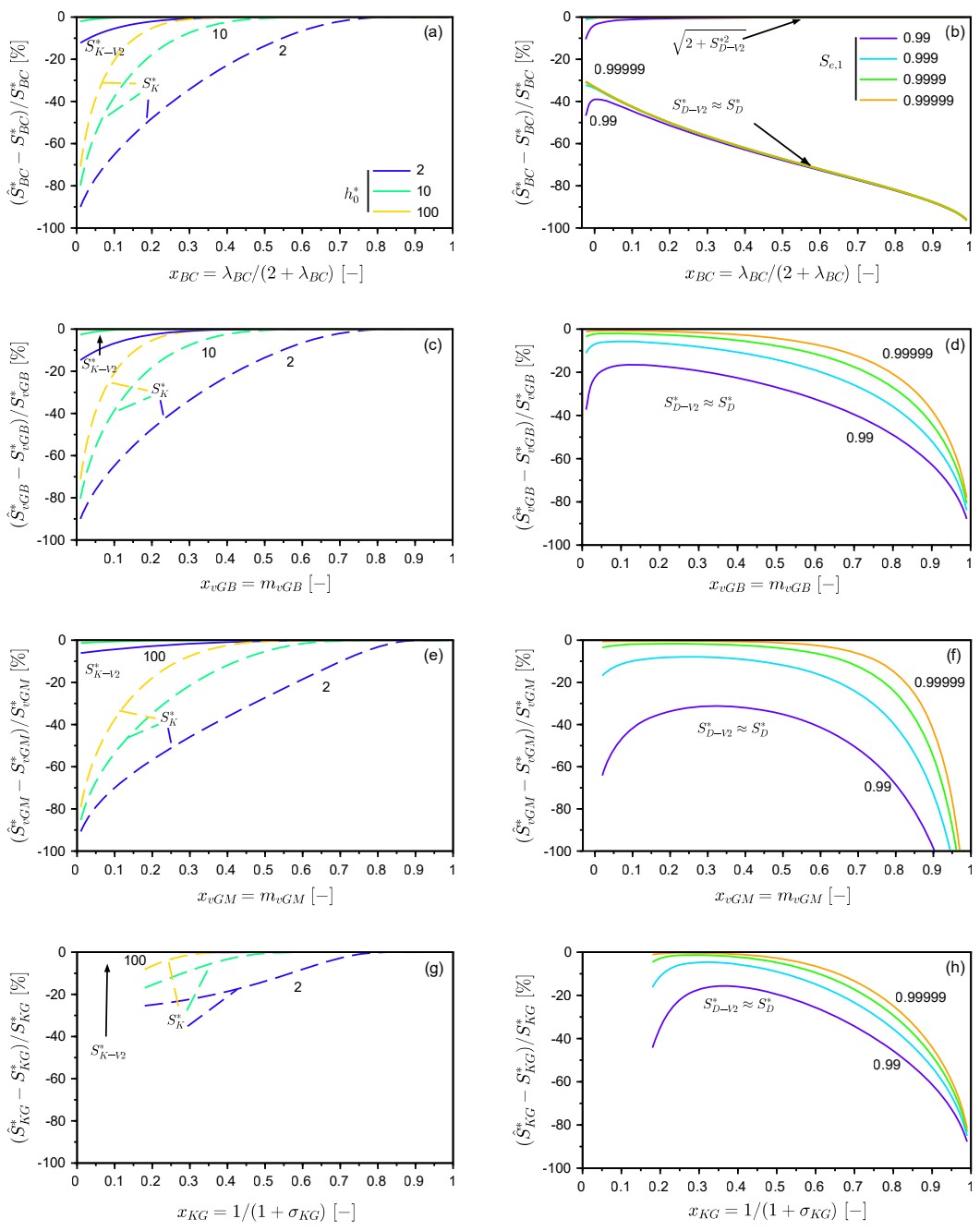

**Figure 6.** Relative errors of estimators $S_{\mathrm{D}}^*$ and $S_{\mathrm{D-V2}}^*$ (right) and $S_{\mathrm{K}}^*$ and $S_{\mathrm{K-V2}}^*$ (left) for the four selected hydraulic models, i.e., BC, vGB, vGM, and KG models as a function of the WR shape indexes.

For the BC model, the estimators $S_\mathrm{D}^*$ and $S_{\mathrm{D}-\mathrm{V2}}^*$ provide poor estimates under all circumstances, since they miss the saturated part of sorptivity, as explained above (Figure 6b, $S_\mathrm{D}^* \approx S_{\mathrm{D}-\mathrm{V2}}^*$)). Adding the saturated part of sorptivity substantially improves the computation, with very accurate estimations (Figure 6b, $\sqrt{(2 + S_\mathrm{D}^{*2})}$). Regarding the estimators $S_\mathrm{K}^*$ and $S_{\mathrm{K}-\mathrm{V2}}^*$ (Figure 6, left column), the estimates improve when the WR shape index converges towards 1. It can be noted that $S_{\mathrm{K}-\mathrm{V2}}^*$ always provide quite accurate predictions with very small relative errors (Figure 6a, c, e, f).

The results obtained in this study also revealed particular behaviors that are specific to different formulations for computing sorptivity, specifically $S_\mathrm{K}^*$, $S_{\mathrm{K}-\mathrm{V2}}^*$, $S_\mathrm{D}^*$, and $S_{\mathrm{D}-\mathrm{V2}}^*$. The approaches that compute sorptivity by integration with regards to $h^*$ (i.e., $S_\mathrm{K}^*$ and $S_{\mathrm{K}-\mathrm{V2}}^*$) have errors that come from the choice of the lower integration boundary $h_0^*$ and from the value of $S_{e,0}$ in the integrand. Solutions that compute sorptivity by integration with regards to $S_e$ (i.e., $S_\mathrm{D}^*$ and $S_{\mathrm{D}-\mathrm{V2}}^*$) have more error associated with the omission of the saturated part, as well as slow convergence of the integration procedure when dealing with

infinite diffusivity values. Among the usual procedures, the use of $S_\mathrm{K}^*$ is better than $S_\mathrm{D}^*$, but the use of $S_{\mathrm{K}-\mathrm{V2}}^*$ remains the best one. Indeed, $S_{\mathrm{K}-\mathrm{V2}}^*$ does not miss the saturated part of sorptivity and provides much lower discrepancies than the other estimators regardless of the selected hydraulic model (Figure 6). At the same time, the adaptation of the integrand, by fixing the values of the initial saturation degree, $S_{e,0}$, to its final values, substantially improves the quality of the estimator. This formulation thus always provides estimates with $|E_\mathrm{r}| < 10\%$, regardless of the selected hydraulic model and the value of shape

index. However, these errors are still much larger than those obtained with the mixed formulation, $S_\mathrm{M}^*$. Consequently, the use of the mixed formulation, $S_\mathrm{M}^*$, should be favored as far as possible.

### 3.4 Computation of sorptivity and accuracy of water infiltration models based on sorptivity

In the following, we illustrate the need to use accurate sorptivity estimates and adequate models for the accurate modeling of water infiltration into soils. We consider the case of one-dimensional (1D) water infiltration into soils for a loamy soil as

a function of initial conditions, i.e., $h_\mathrm{i}$, and final conditions, i.e., $h_\mathrm{f}$. In addition to the illustration of the practical use of the proposed mixed formulation, $S_\mathrm{M}$, we demonstrate that (i) $S_\mathrm{M}$ provides better estimates for sorptivity than the usual estimator, $S_\mathrm{D}$, because it never omits the saturated part of sorptivity, (ii) this improvement avoids significant misestimations of sorptivity and related errors for the modeling of the cumulative infiltration into soils, and (iii) the need to use the correct analytical model that accounts for positive water pressure head at surface. Note that, for the sake of clarity, we compare $S_\mathrm{M}$ to $S_\mathrm{D}$, only, with no

comparison with the other methods since $S_{\mathrm{D}-\mathrm{V2}}$ gives similar results to $S_\mathrm{D}$ and since $S_\mathrm{K}$ and $S_{\mathrm{K}-\mathrm{V2}}$ do not miss the saturated part of sorptivity and involve much lower errors. Lastly, the impact of the selection of the hydraulic models for the description of the soils hydraulic function on the computation of sorptivity and on related modeling of the cumulative infiltration is also discussed.

### 3.4.1 Illustrative example of computation of sorptivity and dependency upon initial and final conditions

The studied synthetic loamy soil was defined by Carsel and Parrish (1988) considering the vGM hydraulic model (Eqs. 28) with the hydraulic parameters tabulated in Table 2 (column "vGM"). In addition, we used the BC and KG models (Eqs. 26 and Eqs. 29), with the parameters tabulated in Table 2 (column "BC" and "KG"). Related hydraulic parameters were optimized by

fitting BC and KG models to the vGM model and are available in HYDRUS software suite (Radcliffe and Simunek, 2018). Fig. 7 depicts the water retention curve (Fig. 7a) and unsaturated hydraulic conductivity (Fig. 7b) and demonstrates the proper alignment of BC and KG models on the vGM model, despite the slightly greater discrepancy for the BC model close to saturation for the hydraulic conductivity (Fig. 7b).

**Table 2.** Hydraulic parameters, i.e., values of the residual and saturated water contents, $\theta_r$ and $\theta_s$, the scale parameter for water pressure head $h_g$, the air-entry water pressure head $h_a$, the saturated hydraulic conductivity $K_s$, and the shape parameters involved in the hydraulic models used for the description of the water retention and the hydraulic conductivity curves, i.e., Eqs.(28), (26), and (29).

| | vGM | BC | KG |
|---|---|---|---|
| $\theta_r$ | 0.078 | 0.078 | 0.078 |
| $\theta_s$ | 0.43 | 0.43 | 0.43 |
| $h_g$ [mm] | -277.8 | -111.5 | -1018 |
| $h_a$ [mm] | 0 | -111.5 | 0 |
| $K_s$ [mm s$^{-1}$] | $2.288\,10^{-3}$ | $3.667\,10^{-3}$ | $2.288\,10^{-3}$ |
| shape parameter | $n_{\mathrm{vGM}} = 1.56$ | $\lambda_{\mathrm{BC}} = 0.34$ | $\sigma_{\mathrm{KG}} = 1.997$ |
| tortuosity parameter | $l_{\mathrm{vGM}} = 1/2$ | $p_{\mathrm{BC}} = 1$ | $l_{\mathrm{KG}} = 1/2$ |

At first, we illustrate the use of the mixed formulation, $S_{\mathrm{M}}$, for the computation of sorptivity for a final water pressure head at surface of $h_{\mathrm{f}} = -150\,\mathrm{mm}$ and an initial water pressure head of $h_{\mathrm{i}} = -10^4\,\mathrm{mm} = -10\,\mathrm{m}$. This initial condition corresponds to the isostatic water pressure head in equilibrium with a water table positioned at 10 m below the ground. The value of $-150\,\mathrm{mm}$, considered at surface, is often used for water infiltration with tension disk experiments to deactivate the macropores and infiltrate only into the matrix (e.g., Timlin et al., 1994; Malone et al., 2004; Lassabatere et al., 2014). Practically, the computation of dimensional sorptivity with the proposed procedure $S_{\mathrm{M}}$ can be performed using the codes developed for this paper and deposited on Zenodo (Lassabatere, 2022). Alternatively, computations may be performed as follows:

1. Compute the initial and final saturation degrees, $S_{\mathrm{e,i}} = S_{\mathrm{e}}(h_{\mathrm{i}})$ and $S_{\mathrm{e,f}} = S_{\mathrm{e}}(h_{\mathrm{f}})$,

2. Compute the intermediate water pressure head, $h_{\mathrm{c}}^* = -\min\left(\left|h^*\left(\frac{S_{\mathrm{e,i}}+S_{\mathrm{e,f}}}{2}\right)\right|, 10^z\right)$ with $z \in \{-2, -1, 0, 1, 2\}$

3. Compute the intermediate saturation degree from the intermediate water pressure head, $S_{\mathrm{e,c}} = S_{\mathrm{e}}(h_{\mathrm{c}}^*)$

4. Integrate the lower part of the squared sorptivity $\mathrm{A} = \int_{S_{\mathrm{e,i}}}^{S_{\mathrm{e,c}}} (S_{\mathrm{e,f}} + S_{\mathrm{e}} - 2S_{\mathrm{e,i}}) D^*(S_{\mathrm{e}}) dS_{\mathrm{e}}$

5. Integrate the upper part of the squared sorptivity $\mathrm{B} = \int_{h_{\mathrm{c}}^*}^{h_{\mathrm{f}}^*} (S_{\mathrm{e,f}} + S_{\mathrm{e}}(h^*) - 2S_{\mathrm{e,i}}) K_{\mathrm{r}}(h^*) dh^*$

6. Combine the two parts to compute the mixed formulation $S_{\mathrm{M}}^* = \sqrt{\mathrm{A} + \mathrm{B}}$

7. Upscale by multiplying the scaled sorptivity, $S_{\mathrm{M}} = \sqrt{|h_g| K_{\mathrm{s}} (\theta_{\mathrm{s}} - \theta_{\mathrm{r}})}\, S_{\mathrm{M}}^*$

For the studied case, with $h_{\mathrm{i}} = -10\,\mathrm{m}$ and $h_{\mathrm{f}} = -150\,\mathrm{mm}$, it leads to the following results:

1. The related scaled water pressure head are $h_i^* = -36$ and $h_f^* = -0.54$, with related saturation degrees of $S_{e,i} = 0.134$ and $S_{e,f} = 0.89$.

2. The intermediate water pressure head takes the value of $h_c^* = -1$, which corresponds to the maximum between $h^*\left(\frac{S_{e,i}+S_{e,f}}{2}\right) = -2.96$ and $-10^0 = -1$.

3. The corresponding saturation degree takes the value of $S_{e,c}^* = 0.780$

4. The computation of the lower part of the squared sorptivity gives the value of $A = 0.0344$

5. The computation of the upper part of the squared sorptivity gives the value of $B = 0.0515$

6. The combination of the two parts gives the scaled sorptivity: $S_M^* = 0.2931$

7. The upscaling finalizes the computation of sorptivity: $S_M = 0.156\,\mathrm{mm\,s}^{-1/2}$

Then, we computed sorptivity with the mixed formulation, $S_M$, and the regular estimator, $S_D$, as a function of initial water pressure head, $h_i$, for the water pressure head at surface of $h_f = -150\,\mathrm{mm}$ (Fig. 7c) and as a function of final water pressure head, $h_f$, for an initial water pressure head of $h_i = -10^4\,\mathrm{mm} = -10\,\mathrm{m}$ (Fig. 7d). We considered three different models (vGM, BC, and KG). We first analyzed the values of sorptivity as estimated by the mixed formulation $S_M$. The values of sorptivity decrease with the initial water pressure head, $h_i$, (Fig. 7c) and increase with the final water pressure head, $h_f$, (Fig. 7d), regardless of the chosen hydraulic model, vGM, BC, or KG. This trend was expected since the integrand is always positive, and the integration of positive functions decreases with regards to its lower boundary and increases with regards to its upper boundary. Those trends have already been documented by many authors (e.g. Stewart et al., 2013). The values of sorptivity were also computed with the regular estimator $S_D$ (Fig. 7c-d, dashed lines). In most cases, the two estimators, $S_M$ and $S_D$ provided the same estimations, except when the final pressure head was higher than the air-entry water pressure head, $h_f \geq h_a$. In this case, as explained above, $S_D$ missed the saturated part of the sorptivity, which explains why the values predicted with $S_D$ no longer increase with $h_f$ but remain equal to the unsaturated part of sorptivity: $S_D(\theta_i, \theta(h_f \geq h_a)) = S_D(\theta_i, \theta_s) = S_u(\theta_i, \theta_s)$ (Fig. 7d, plateaus with dashed lines). Note for vGM and KG models, the air-entry water pressure head is zero, then the plateaus begin at $h_f = 0$, whereas the air-entry water pressure head is lower than zero for the BC model, $h_a = -111.5\,\mathrm{mm}$, inducing an earlier beginning of the plateau (Fig. 7d, BC versus vGM and KG).

Another interesting point is the difference between the three models. The BC model exhibits much larger sorptivity values, followed by the vGM model, and then the KG model with the lowest values. However, the three models are meant to represent the same water retention and hydraulic conductivity functions, i.e., the same soil. That result shows that the selection of the hydraulic model for the water retention and hydraulic conductivity functions impacts significantly the value of sorptivity, even when the models are fitted to the same data and represent the same soil. This point was already raised by Lassabatere et al. (2021). This feature is counter-intuitive since the use of close hydraulic models is expected to provide close values of sorptivity. Such a discrepancy may result from the different mathematical behaviors of the three different hydraulic models close to saturation. A closer look at the hydraulic functions shows that the unsaturated hydraulic conductivity differs close to

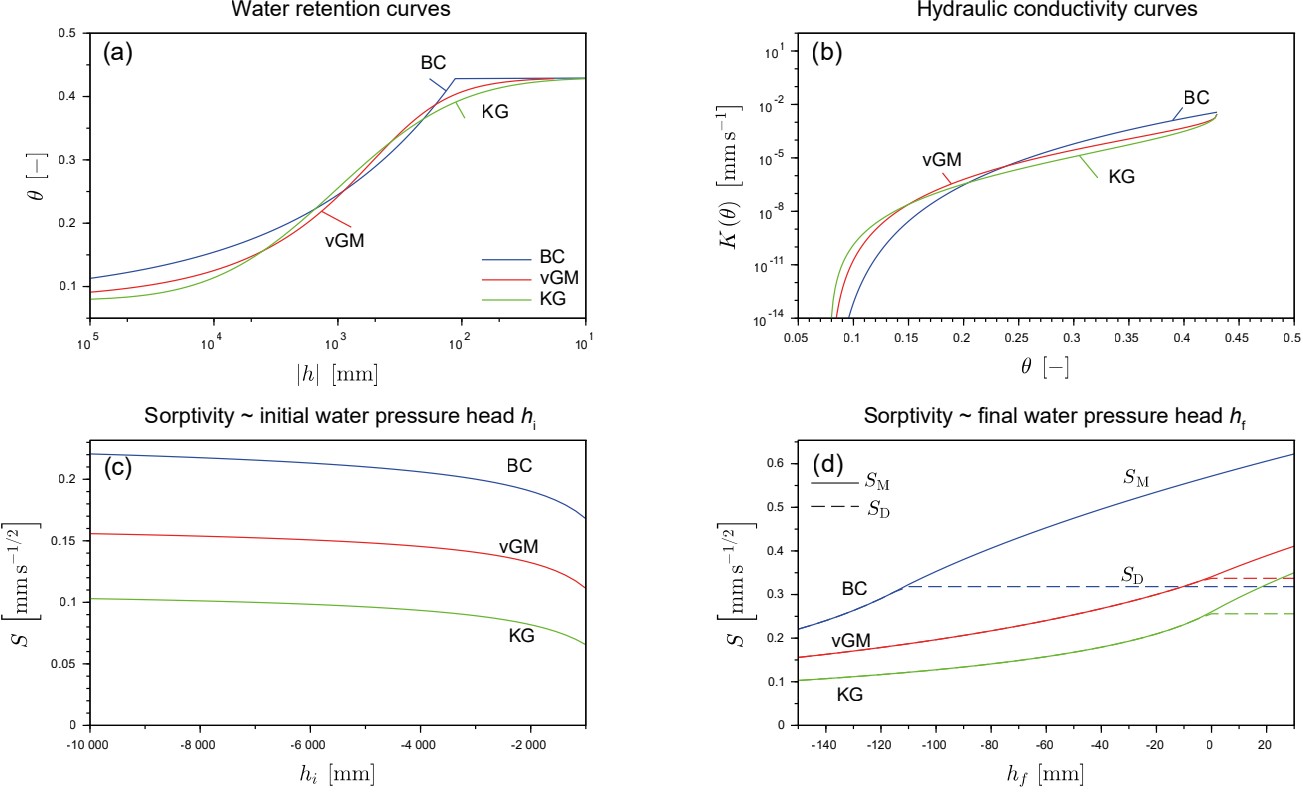

**Figure 7.** Water retention (a) and hydraulic conductivity (b) curves described by the three sets of hydraulic models (BC, vGM, and KG), sorptivity as function of initial (c) and final (d) water pressure heads with the sorptivity estimated by the proposed mixed formulation $S_M$ and the regular model $S_D$.

saturation with the following ranking: BC » vGM » KG (Fig. 7b), which may explain the observed discrepancy between values

of sorptivity. We suspect that the ranking of the values of sorptivity between models reflects the ordering between unsaturated hydraulic conductivity close to saturation. Stewart and Abou Najm (2018) also attributed higher sorptivity values with the use of BC function to the extra space under the $K(h)$ curve.

### 3.4.2    Impact of misestimations of sorptivity on cumulative infiltrations

Given the dependence of sorptivity on the choice of the estimator and the hydraulic model, we expected some implications for

water infiltration. To demonstrate this, we modeled water infiltration into the studied loamy soil, with a 30 mm water pressure head at surface ($h_f = 30\,\mathrm{mm}$) and an initial water pressure head of -10 m ($h_i = -10^4\,\mathrm{mm}$). To model water infiltration, we used the analytical model developed for $h_f \leq h_a$ by Haverkamp et al. (1994) and its extension to $h_f > h_a$ developed by Haverkamp et al. (1990). The first model, referred to as the quasi-exact implicit (QEI) model by many authors (e.g. Fernández-Gálvez

et al., 2019), reads as follows:

$$\frac{2\Delta K^2}{S^2}t = \frac{1}{1-\beta}\left(\frac{2\Delta K}{S^2}\left(I_{1\mathrm{D}} - K_{\mathrm{i}}t\right) - \ln\left(\frac{e^{\frac{2\beta\frac{\Delta K}{S^2}}(I_{1\mathrm{D}}-K_{\mathrm{i}}t)} + \beta - 1}{\beta}\right)\right) \tag{40}$$

where $\Delta K$ stands for the difference between final and initial values of hydraulic conductivity $\Delta K = K_{\mathrm{f}} - K_{\mathrm{i}}$, and $\beta$ stands for an infiltration constant, usually fixed at 0.6. Varado et al. (2006) followed by Lassabatere et al. (2009) suggested introducing the following scaling procedure for time $t$ and cumulative infiltrations $I$:

$$\begin{cases} \gamma_{\mathrm{I}} = \frac{S^2}{2\Delta K} & I = \gamma_{\mathrm{I}}I^* + K_{\mathrm{i}}t \\ \gamma_{\mathrm{t}} = \frac{S^2}{2\Delta K^2} & t = \gamma_{\mathrm{t}}t^* \end{cases} \tag{41}$$

These scaling equations lead to the following relationship between the scaled and dimensional versions of the QEI model (Lassabatere et al., 2009):

$$I_{\mathrm{QEI}}(t) = \gamma_{\mathrm{I}}I^*_{\mathrm{QEI}}\left(\frac{t}{\gamma_{\mathrm{t}}}\right) + K_{\mathrm{i}}t \tag{42}$$

with the following equations for the scaled QEI model, $I^*_{\mathrm{QEI}}(t^*)$:

$$t^* = \frac{1}{1-\beta}\left(I^* - \ln\left(\frac{e^{\beta I^*} + \beta - 1}{\beta}\right)\right) \tag{43}$$

Note that in both equations Eq. (40) and Eq. (43), the cumulative infiltration is defined implicitly, time being defined as a function of the cumulative infiltration. On the same basis, Ross et al. (1996) scaled the model developed by Haverkamp et al. (1990) for the case of $h_{\mathrm{f}} > h_{\mathrm{a}}$, leading to:

$$I^* = \frac{\sigma}{q^* - 1} + \frac{1-\sigma}{\beta}\ln\left(1 + \frac{\beta}{q^* - 1}\right) \tag{44}$$

$$t^* = \frac{1-\sigma}{\beta(1-\beta)}\ln\left(1 + \frac{\beta}{q^* - 1}\right) + \frac{\sigma}{q^* - 1} - \frac{1-\sigma\beta}{1-\beta}\ln\left(1 + \frac{1}{q^* - 1}\right) \tag{45}$$

where $q^* = \frac{q}{\gamma_{\mathrm{q}}}$ is the scaled infiltration rate and $\gamma_{\mathrm{q}} = \Delta K$. In this case, there is no direct relationship between the time $t^*$ and the cumulative infiltration $I^*$, but two relationships with the first relating time and the infiltration rate, $t^* = t^*(q^*)$, defined by Eq. (45), and the second relating the cumulative infiltration and the infiltration rate, $I^* = I^*(q^*)$, defined by Eq. (44). To compute the cumulative infiltration as a function of time, the infiltration rate $q^*$ must be retrieved as the root of equation from Eq. (45) before being inserted into Eq. (44) to define the scaled infiltration as a function of the scale time, $I^*_{\mathrm{QEI}_{\mathrm{ext}}}(t^*)$. In the following, this relationship is referred to as the extended version of the QEI model and noted QEI_ext model. This set of scaled models, Eq. (44-45), must be upscaled considering the same scaling factors, as defined in Eqs. (41) with the additional parameter, $\sigma$, which quantifies the relative contribution of the saturated part of sorptivity to the total sorptivity:

$$\sigma = \frac{2\left(\theta_{\mathrm{f}} - \theta_{\mathrm{i}}\right)K_{\mathrm{f}}\left(h_{\mathrm{f}} - h_{\mathrm{a}}\right)}{S^2} \tag{46}$$

The QEI_ext model addresses all cases. When $h_{\mathrm{f}} \le h_{\mathrm{a}}$, the saturated part of sorptivity is null leading to $\sigma = 0$, and the QEI_ext model reduces to the regular QEI model. When $h_{\mathrm{f}} > h_{\mathrm{a}}$, the regular QEI model is no longer valid and the extended QEI_ext

model should be used instead. In other words, QEI_ext model is always valid and should be preferred to the QEI model in any case. In the following example, we consider that two different errors may arise: (i) the wrong choice of estimator for the sorptivity, and (ii) the wrong choice of the model. In the following, we investigate the effect of those two errors.

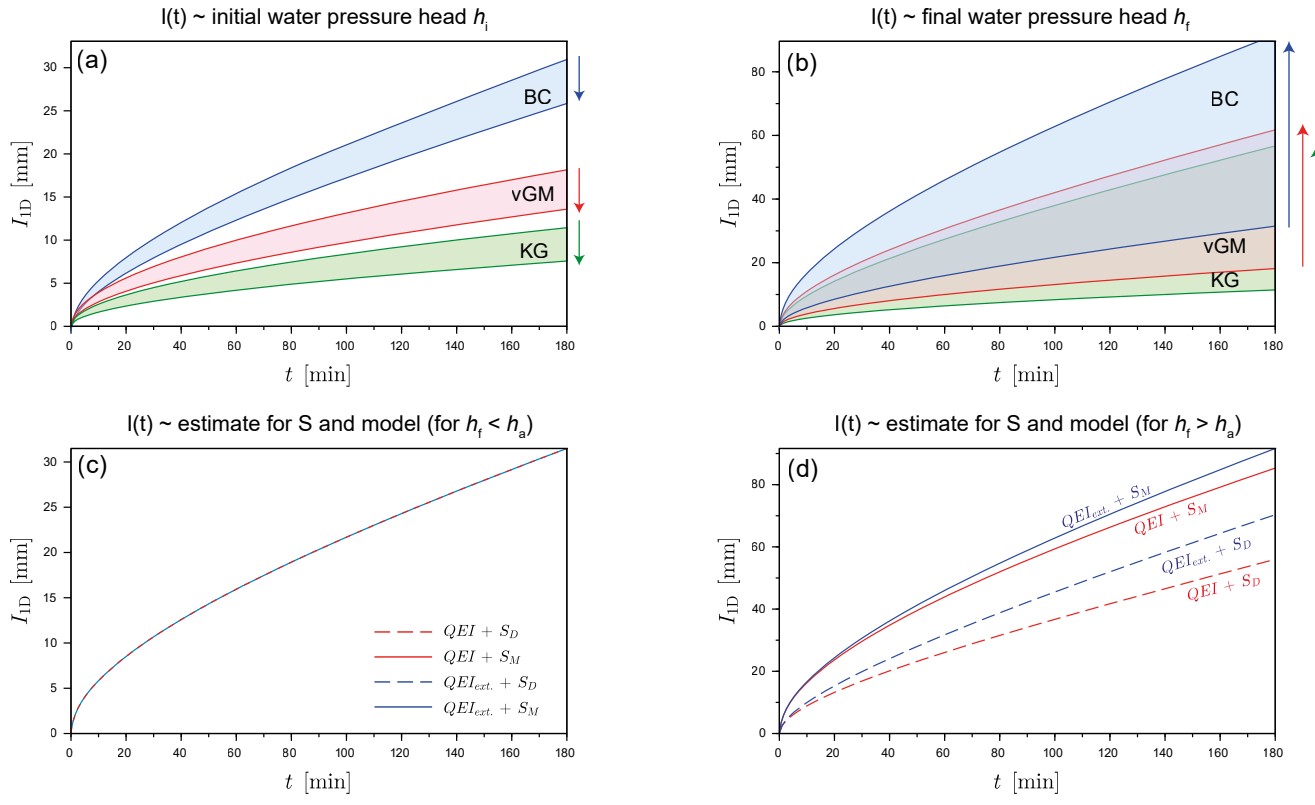

**Figure 8.** Cumulative infiltrations as function of initial (a) and final (b) water pressure heads, computed with the correct estimate for sorptivity ($S_M$) and the right model, and illustration of the dependency of cumulative infiltration on the model selection (QEI model versus extended QEI_ext model) and on the selected for the sorptivity ($S_M$ versus $S_D$) for (c) the case of unsaturated conditions imposed at surface $h_f \leq h_a$ (BC model, $h_i = -10$ m and $h_f = -150$ mm) and (d) the case of saturated conditions, $h_f \geq h_a$ (BC model, $h_i = -10$ m and $h_f = 30$ mm).

Before investigating the impact of errors, we evaluated the dependence of the cumulative infiltrations as a function of initial and final conditions. For this analysis, we use the mixed formulation for sorptivity and the extended QEI_ext model and obtained the results depicted in Fig. (8a-b). Cumulative infiltration decreases with the initial water pressure head $h_i$ (Fig. 8a, arrows downwards) and increases with the final water pressure head $h_f$ (Fig. 8b, arrows upwards), as a consequence of the variations of sorptivity as a function of $h_i$ and $h_f$ (Fig. 7c-d). It can be noted that the impact of the water pressure head at
surface $h_f$ is much more important, with much wider curve beams (Fig. 7d versus c, see shaded areas). Again, these trends are in line with previous studies (e.g., Lassabatere et al., 2009, 2014; Angulo-Jaramillo et al., 2016). The driver of water infiltration into the soil is the hydraulic gradient and thus the difference between the final and the initial water pressure head. The higher

the final water pressure head and the lower the initial water infiltration, the higher the hydraulic gradient and thus the higher the cumulative infiltration. As for the sorptivity, the selection of hydraulic model for the water retention and hydraulic conductivity

functions drastically impacts the results, with the same ordering than for sorptivity, i.e., largest cumulative infiltration for BC, then vGM, and lastly KG model. As a consequence, the fit of specific hydraulic models on the same water retention and hydraulic conductivity functions may lead to contrasting results in terms of water infiltration. To the authors' knowledge, this is the first time that the influence of the choice of the hydraulic model on sorptivity estimation and water infiltration has been so clearly demonstrated.

As stated above, either a wrong estimation of sorptivity or a wrong choice of the analytical model (QEI instead of QEI_ext model) may lead to erroneous estimates of the cumulative infiltration. The four combinations of the two models and the two estimates of sorptivity were used to model water infiltration for the case of an initial water pressure head of $h_\mathrm{i} = -10^4\,\mathrm{mm}$ and a final water pressure head of $h_\mathrm{f} = -150\,\mathrm{mm}$ (Fig. 8c), and for the case of the same initial water pressure head, $h_\mathrm{i} = -10^4\,\mathrm{mm}$, but a positive surface water pressure head $h_\mathrm{f} = 30\,\mathrm{mm}$ (Fig. 8d). For the first case, there was no difference at all (Fig. 8c). Both

$S_\mathrm{D}$ and $S_\mathrm{M}$ accurately quantified the sorptivity and the two models, QEI and QEI_ext were similar. In contrast, for the second case, the difference were quite significant. The use of the more appropriate estimator, i.e., $S_\mathrm{M}$ instead of $S_\mathrm{D}$, increased the amount of modeled cumulative infiltration, regardless of the selected model. In addition to that, the use of the correct model, i.e., the QEI_ext instead of the QEI model substantially increased the cumulative infiltration, with a slightly lower increase in comparison to that brought by the choice of $S_\mathrm{M}$ for the computation of sorptivity. These results demonstrate the importance of

using the proper estimator for sorptivity in combination with the best possible infiltration model.

## 4 Conclusions

The proper calculation of sorptivity is crucial to model accurately water infiltration into soils. However, in some cases, e.g., when the initial state is very dry or the final state corresponds to saturated conditions, the numerical computation of sorptivity using Eq. (4) or Eq. (5) from the hydraulic functions may be a source of numerical errors and difficulties. Indeed, the integration

procedure typically used involves either an infinite boundary or unbounded integrands. Many previous studies had attempted to alleviate these problems by fixing arbitrarily finite limits for the integration interval. In this study, we investigated the accuracy of these approaches and demonstrated the potentially massive mis-estimation of sorptivity that is possible when these arbitrary corrections are used. To alleviate those problems, we proposed a mixed formulation that was validated against analytical expressions of sorptivity for specific hydraulic models. The proposed mixed-formulation proved highly accurate

for all hydraulic models and shape parameters tested, with negligible relative errors ($< 10^{-7}$). Conversely, the use of regular estimates for sorptivity lead to large under-estimates in many instances, in particular when the final water pressure head exceeds the air-entry water pressure head.

This study demonstrates that, through the use of the new mixed formulation, it is possible to compute sorptivity easily and very accurately. The proposed formulation presents a very practical tool that may be applied for any type of hydraulic model and

any value of initial and final water pressure heads and water contents. The proposed approach allows sorptivity to be computed

in all cases, thus improving the modeling of water infiltration into soils and the estimation of soil hydraulic properties. In addition, we used the proposed formulation to investigate the sensitivity of sorptivity to initial and final water pressure heads and to the choice of the hydraulic model chosen to quantify the water retention and unsaturated hydraulic curves. This analysis clearly demonstrated that sorptivity increases with the final water pressure head and decreases with the initial water pressure. We also showed that a proper estimate of sorptivity is crucial with regards to the modeling of water infiltration into soils, and that the selection of the model for the hydraulic functions drastically impacts the computation of sorptivity and, consequently, the final amounts of cumulative infiltrations.

*Code availability.* Note all computations were done using Scilab free software. The scripts for the computation of all the results and figures presented in this paper, and in particular the script for the proposed mixed formulation can be downloaded online: https://zenodo.org/record/5789111 (Lassabatere, 2022).

*Author contributions.* L.L. established the question, performed the analytical developments, computed the numerical results, and provided the first draft of the manuscript. P.-E.P. confirmed the analytical and numerical developments and wrote with L.L. the first draft. D.Y., B.L., D.M.-F., S.d.P. and M.R. verified parts of the numerical computations. J. P. and J. F.-G. helped for the use of the Kosugi model. R.D.S. and M.A.N. helped for the editing, the layout of the manuscript and the presentation of the results. All the authors contributed to the editing of the manuscript.

*Competing interests.* No competing interest to declare.

*Acknowledgements.* This work was performed within the INFILTRON project supported by the French National Research Agency (ANR-17-CE04-010).

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
