# Peer review of "Mixed formulation for an easy and robust numerical computation of sorptivity"

_Hydrology and Earth System Sciences, 2021_

## Author Comment (AC1)

Hydrol. Earth Syst. Sci. Discuss., referee comment RC1
https://doi.org/10.5194/hess-2021-633-RC1, 2022

[Figure]

**Authors response to comment on hess-2021-633**

Anonymous Referee #1

Referee comment on "Mixed formulation for an easy and robust numerical computation of sorptivity" by Laurent Lassabatere et al., Hydrol. Earth Syst. Sci. Discuss., https://doi.org/10.5194/hess-2021-633-RC1, 2022

**Manuscript: hess-2021-633**

**Title: Mixed formulation for an easy and robust numerical computation of sorptivity**

Dear editor and authors. I read carefully the manuscript. The article is quite interesting and is well written and organized. The authors propose a new mixed formulation that scales sorptivity. The topic is relevant and gives valuable information about sorptivity. This hydraulic property is a key parameter, and there is a lack of information about it, especially in terms of the dependency with the soil water content. For these reasons, the manuscript fits into the journal's scope and would be relevant to the readers of Hydrology and Earth Systems Science. Additionally, this manuscript complements very well the previous study of Lassabatere et al. (2021), published in this journal. However, there is one major concern that must be addressed before publication. Soil sorptivity is a function of the initial and final soil water content. Since sorptivity is an expression of the capillarity forces, the highest value corresponds to the dry condition (h=-∞) and decreases as the soil water content increases (in dry condition, the capillary forces domains the process, while near saturation there is no expression of capillarity). The authors included in the manuscript the Figures 1 and 2 in order to show an example of the mixed formulations. In these figures, the y axis corresponds to the sorptivity (estimated with the new mixed formulation function) and the x axis corresponds to the soil water content or water pressure head (h). The behavior of this function is the opposite to the expected one. This issue should be addressed in the manuscript. Additionally, the inclusion of hypothesis and objective will improve the manuscript. Also, it would be very interesting to include some figures with the sorptivity values as function of soil water content and water pressure head, calculated with the new mixed formulation. Below, I mention more detailed comments. I'm not English native speaker, then I will not correct language issues.

**Authors:** The authors warmly thank the reviewer for their careful review of the paper and positive comments on the proposed study. As required by the reviewer, the authors will provide a new version with a more straightforward presentation of the paper's objectives before the theory section and a strengthened conclusion. The revised version will also include an additional section at the end of the manuscript to describe the proposed mixed formulation's practical use and provide a sensitivity analysis of sorptivity as a function of the initial and final water contents.

Besides, the authors would like to clarify figures 1 and 2 and answer to the apparent inconsistency raised by the reviewer. In these figures, we plotted the integrand to be integrated for the computation of sorptivity. This integrand defines an increasing function with regards to the integration variable, θ. The computation of sorptivity involves the

integration of the integrand between the initial and the final water contents. Given that the integrand is positive, sorptivity defines a decreasing function with regards to the initial water content and an increasing function with regards to the final water content. This point is in full accordance with the reviewer's statement on sorptivity variation with water content. Furthermore, this result does not contradict figures 1 and 2. The reviewer might have gotten and thought that sorptivity was plotted instead of the integrand, which made him believe that our plots were inaccurate.

Detailed comments:

L 12-13: the first two sentences are exactly the same than the two first sentences of Lassabatere et al. (2021). Please modify.

**Authors:** The authors apologize and will change the two first sentences.

L 20: Equation (1): Please add more information about this equation. I couldn't find the same expression in Parlange (1975).

**Authors:** The authors will clarify the relationship between Parlange (1975) study and the integral expression defined by Equation (1).

L 21: initial and final water contents of the soil or the water source? More detailed information about the relationship between sorptivity and water content is needed.

**Authors:** The reviewer is correct. By final water content or water pressure head, the authors mean the conditions imposed at the surface (i.e., the water source). The revised manuscript will be clarified in this regard. The authors will add a section on the practical computation of sorptivity and its link to the initial and final water contents.

L 24: I couldn't find the same expression in Ross et al. (1996). Please give more details about the construction of this equation.

**Authors:** The reviewer is correct. This part is unclear and will be rewritten in the revised manuscript. Ross et al. (1996) did not clearly write the equation as it is mentioned in our study, even if he suggested in the text to express variables as water pressure heads. We will refer to more appropriate citations.

L 240-246: Please use these ideas to build hypothesis and objectives, and include them in the Introduction section.

**Authors:** The reviewer is correct. This part will be used to define clearly the objectives of the study at the end of the introduction of the paper.

L 363-371: this is not a conclusion. The inclusion of an explicit hypothesis will improve this section.

**Authors:** The reviewer is correct. The conclusion will be strengthened in the revised version of the paper, with more details on the taking-home message of our study and more insight into perspectives and further works.

---

## Author Response (AR1)

Hydrol. Earth Syst. Sci. Discuss., referee comment RC1
https://doi.org/10.5194/hess-2021-633-RC1, 2022

[Figure]

**Authors response to the editor and reviewers' comments on hess-2021-633**

**Editor:**

Dear Authors,
Allowing for the fairly positive evaluations of your original submission and your preliminary responses, your study passes on the subsequent journal step and you are invited please upload a revised version. Please upload also a more structured point-by-point reply that comprises all of the comments received so far and also highlights the main changes. Should you disagree with some comments of the reviewers, please explain why clearly in the rebuttal. An additional referee step might be advised.

**Authors:**

Dear Editor, we have done our best to address both reviewers' comments. The first reviewer highlighted the lack of clarity of the objectives and the novelty of the input of the proposed study. In addition, some comments point to misunderstanding, revealing the lack of clarity in some parts of the manuscript (in particular in the theory section). We have addressed all of the reviewers' suggestions and significantly modified the manuscript to comply with the reviewers' suggestions.

The new manuscript now includes the following novelties:

- An improved presentation of the paper's objectives and goals, with additional objectives compared to the previous versions. The new objectives are related to the new section demonstrating the need for accurate estimates of sorptivity (lines 60-67 and 78-80 of the marked revised manuscript).

- A brand-new section (Section 3.4, pp. 22-28, marked revised manuscript) on the gain in accuracy of the proposed mixed formulation and its benefits regarding the modeling of water infiltration into soils. This new part gives further insight into the sensitivity of sorptivity with regard to initial and final conditions and into the choice of the model for water retention and hydraulic conductivity functions. It also investigates the use of sorptivity for modeling water infiltration into soils and the impact of erroneous estimates of sorptivity on the quality of the prediction of water infiltration. This part strengthens the novelty of the paper since no study has previously investigated the dependency of sorptivity estimation on the mathematical procedure and related consequences on the quality of predictions of water infiltration into soils.

- An improved conclusion that reminds the reader of the main taking-home messages and the related implications with regards to the more general topic of predicting water infiltration into soils (lines 535-537, 543-552, marked revised manuscript). In particular, we insist that an accurate estimation of sorptivity is crucial with regard to the quality of the model of water infiltration into soils. Misestimation of sorptivity may lead to a significant misprediction of cumulative infiltrations. Besides, we demonstrated that even when sorptivity was appropriately estimated, the choice of the mathematical functions used to describe water retention and hydraulic conductivity functions played a significant role, by increasing sorptivity and cumulative infiltrations by more than two times, which is worth mentioning.

In addition to those main features, we made these additional improvements:

- All the figures were improved with an increase in the font size of labels and improvement of lines and legends,
- In particular, we improved Fig. 2, which generated some misunderstandings (see R1 reviewer's comments). This figure was replaced with two different figures for each case (dry initial condition versus close to saturation). In addition, the new figures were worked to improve clarity.
- Section 3.4 gives more details on the application of the mixed formulation, which may help anyone with the use of the mixed formulation.
- The code available on Zotero was updated.

The authors hope that the revised version of the manuscript will meet the standards of the journal and will be considered for publication in HESS.

**Anonymous Referee #1**

Referee comment on "Mixed formulation for an easy and robust numerical computation of sorptivity" by Laurent Lassabatere et al., Hydrol. Earth Syst. Sci. Discuss., https://doi.org/10.5194/hess-2021-633-RC1, 2022

Manuscript: hess-2021-633

Title: Mixed formulation for an easy and robust numerical computation of sorptivity

Dear editor and authors. I read carefully the manuscript. The article is quite interesting and is well written and organized. The authors propose a new mixed formulation that scales sorptivity. The topic is relevant and gives valuable information about sorptivity. Thishydraulic property is a key parameter, and there is a lack of information about it, especially in terms of the dependency with the soil water content. For these reasons, the manuscript fits into the journal's scope and would be relevant to the readers of Hydrology and Earth Systems Science. Additionally, this manuscript complements very well the previous study of Lassabatere et al. (2021), published in this journal. However, there is one major concern that must be addressed before publication. Soil sorptivity is a function of the initial and final soil water content. Since sorptivity is an expression of the capillarity forces, the highest value corresponds to the dry condition (h=-∞) and decreases as the soil water content increases (in dry condition, the capillary forces domains the process, while near saturation there is no expression of capillarity). The authors included in the manuscript the Figures 1 and 2 in order to show an example of the mixed formulations. In these figures, the y axis corresponds to the sorptivity (estimated with the new mixed formulation function) and the x axis corresponds to the soil water content or water pressure head (h). The behavior of this function is the opposite to the expected one.This issue should be addressed in the manuscript. Additionally, the inclusion of hypothesis and objective will improve the manuscript. Also, it would be very interesting to include some figures with the sorptivity values as function of soil water content and water pressure head, calculated with the new mixed formulation. Below, I mention more detailed comments. I'm not English native speaker, then I will not correct language issues.

**Authors:** The authors warmly thank the Reviewer for their careful review of the paper and positive comments on the proposed study. As required by the Reviewer, the authors improved the manuscript with a more straightforward presentation of the paper's objectives before the theory section and a strengthened conclusion. The revised version also includes an additional section that investigates the evolution of sorptivity with initial and final water contents as required by the Reviewer, in addition to the impact of sorptivity estimation accuracy on the modeling of water infiltration. The results align with the reviewer's expectations, and show an increase of sorptivity when either the initial condition becomes drier or the final condition becomes wetter (Figure 7, c and d of the new manuscript). Furthermore, we investigated the impact of sorptivity estimation when using the regular versus the proposed mixed formulation

on the results of modeling water infiltration. The different estimates were inserted into the analytical models developed for the computation of 1D cumulative infiltration. Clearly, the choice of the regular methods leads to poor estimates for sorptivity and subsequent improper estimation of cumulative infiltration. Conversely, using the exact mixed formulation ensures a proper estimate of sorptivity and accurate modeling of cumulative infiltration. We also investigated the dependency of sorptivity upon the choice of the hydraulic models for describing the water retention (WR) and hydraulic conductivity (HC) curves. The BC and KG models were fitted to the vGM model as defined for the loamy soils in the soil database by [1]Carsel and Parrish (1988), and computations were performed for the three models. We revealed that using the three hydraulic models leads to significantly different cumulative infiltration values (Figure 8, revised version). This point highlights the need for choosing appropriate models for describing WR-HC functions, even when no clear choice may be made in advance and when the three models provide accurate fits of the same WR and HC curves.

Besides, we also improved the manuscript and some parts that confused Reviewer R1. For instance, Figure 2 led to some misunderstanding, as the Reviewer thought that sorptivity was plotted against the water content or the water pressure head. Instead of the sorptivity, we plotted the integrand, the integration of which gives the sorptivity. This integrand defines an increasing function with regards to the integration variable, $\theta$ or $h$. The computation of sorptivity involves the integration of the integrand between the initial and the final water contents, or initial and final water pressure heads. Given that the integrand is positive, such an integration (i.e., sorptivity) defines a decreasing function with regards to the initial water content (or water pressure head) and an increasing function with regards to the final water content (or water pressure head). This point is in full accordance with the Reviewer's statement on sorptivity. Clearly, these relationships were not explained with sufficient clarity in the previous version. In the revised version, two figures (Figs. 2 and 3) replace the previous Figure 2, and the text was rewritten to clarify statements. We hope the new version is much clearer and satisfies the valid Reviewer comment.

Detailed comments:

L 12-13: the first two sentences are exactly the same than the two first sentences of Lassabatere et al. (2021). Please modify.

**Authors:** Thee sentences were rewritten in the revised manuscript (see Lines 11-16 of the marked revised manuscript).

L 20: Equation (1): Please add more information about this equation. I couldn't find thesame expression in Parlange (1975).

**Authors:** Equation (1) results from the concatenation of the relation between squared sorptivity and the flux concentration function proposed by Philip and Knight (1974), on the one hand, and the formulation proposed by Parlange in 1975 for the flux concentration function, on the other hand. Haverkamp et al. (2006) detailed these relationships as follows:
* * *
[1] Carsel, R.F., Parrish, R.S., 1988. Developing joint probability distributions of soil water retention characteristics. Water Resour. Res. 24, 755–769. https://doi.org/10.1029/WR024i005p00755

$$S_1^2(\theta_1, \theta_0) = 2 \int_{\theta_0}^{\theta_1} \frac{[\theta - \theta_0] D(\theta)}{F(\theta)} d\theta \tag{6.75}$$

Probably the best understanding of the structure of any solution of Equation 6.66 can be based on the analytical expansion technique of Heaslet–Alksne (1961) which was applied by Parlange et al. (1992). A good compromise for the $F$-function which balances accuracy with simplicity (Elrick and Robin, 1981) was given by Parlange (1975):

$$F(\theta) = \frac{2[\theta - \theta_0]}{[\theta_1 + \theta - 2\theta_0]} \tag{6.76}$$

resulting in the sorptivity equation used for many studies (e.g., Ross et al., 1996):

$$S_1^2(\theta_1, \theta_0) = \int_{\theta_0}^{\theta_1} [\theta_1 + \theta - 2\theta_0] D(\theta) d\theta \tag{6.77}$$

*Extracted from: Haverkamp, R., Debionne, D., Viallet, P., Angulo-Jaramillo, R., de Condappa, D., 2006. Soil properties and moisture movement in the unsaturated zone, in: Delleur, J.W. (Ed.), The Handbook of Groundwater Engineering. CRC Press, Indiana, pp. 1–59.*

The expression of the flux concentration function as detailed in (6.76) was proposed by Parlange (1975) in their paper entitled "On solving the flow equation in unsaturated soils by optimization: horizontal infiltration". Their equation [14] defines the optimum flux concentration function as in Eq. (6.76) and related expression for sorptivity, for the case of saturated final water content ($\theta_1 = \theta_s$):

This expression can be simplified (see Appendix) to yield

$$\lambda_0 \approx \left[ \int_0^1 D d\theta + \int_0^1 \theta \, D d\theta \right]^{1/2} / \int_0^1 \theta \, D d\theta \tag{13}$$

which gives the optimal value of $\lambda_0$ associated with the form of $\theta$ given in Eq. [10]. In particular the approximation for the sorptivity is,

$$S = \int_0^1 \phi d\theta \cong \left[ \int_0^1 D d\theta + \int_0^1 \theta \, D d\theta \right]^{1/2} . \tag{14}$$

For instance, for Yolo light clay, this yields

$$10^2 S \approx 1.2490 \tag{15}$$

*Extracted from: Parlange, J.-Y., 1975. On Solving the Flow Equation in Unsaturated Soils by Optimization: Horizontal Infiltration. Soil Science Society Of America Journal 39, 415–418. [Note that in their expression θ refers to the saturation degree.]*

In a subsequent letter to the editor, Parlange provided a clearer expression of the same findings (Parlange, 1975. Determination of Soil Water Diffusivity by Sorptivity Measurements.

Soil Science Society of America Journal 39, 1011–1012), corresponding exactly to Eq. (1) of the original manuscript and Eq. (3) of the revised manuscript:

**Determination of Soil Water Diffusivity by Sorptivity Measurements**

A recent paper by Dirksen (1975) discusses a method for the measurement of soil water diffusivity. The method assumes that the sorptivity, $S$, is related to a weighted mean diffusivity by the equation

$$\int_{\theta_o}^{\theta_1} (\theta - \theta_o)^\gamma D(\theta)\, d\theta = (1 + \gamma)^{-1} (\theta_1 - \theta_o)^{\gamma-1} \Pi\, S^2/4. \quad [1]$$

The form of this equation is such that it becomes exact if the diffusivity $D$ is a constant. $\theta_o$ is the initial water content, $\theta_1$ the water content at the soil surface and $\gamma$ is a constant. Following a numerical test, Dirksen took $\gamma = 0.67$. Considering the usual scatter of absorption experiments, there is no question that Eq. [1] with $\gamma = 0.67$ is quite accurate for most practical purposes. However, rather than choose $\gamma$ by a numerical test it would be more satisfying from a theoretical point of view to take $\gamma$ so that Eq. [1] holds when $D$ approaches a delta function as well, or

$$\gamma = \Pi/2 - 1 \simeq 0.57. \quad [2]$$

This is close enough to the value chosen by Dirksen to be just as accurate in any practical case.

Actually since $D$ varies very rapidly with $\theta$ for all soils, it is worthwhile to look for a relationship between $S$ and the weighted diffusivity which becomes exact when $D$ approaches a delta function, i.e.,

$$\int_{\theta_o}^{\theta_1} (\theta - \theta_o)^\gamma D(\theta)\, d\theta \simeq (\theta_1 - \theta_o)^{\gamma-1} S^2/2. \quad [3]$$

Of course when $\gamma \simeq 0.57$ Eq. [1] and [3] are identical. If one does not require that Eq. [3] should hold in the linear case then $\gamma$ is arbitrary. For instance, when $\gamma = 0$, Dirksen showed with his numerical example that Eq. [1] yields a value of $D$ which is too large by 40% while Eq. [3] yields a value which is too small but only by 10%. This points out the validity of Eq. [3] for real soils.

The best choice for $\gamma$ is now obtained by optimization. It can be shown (Parlange, 1975) that if $D$ varies rapidly near $\theta_1$, then

$$S^2 \simeq \int_{\theta_o}^{\theta_1} (\theta - \theta_o + \theta_1 - \theta_o)\, D(\theta) d\theta. \quad [4]$$

Expanding $(\theta - \theta_o)^\gamma$ in Eq. [3] for $\theta$ in the neighborhood of $\theta_1$ shows that Eq. [3] and [4] are identical to the first two orders when $\gamma = ½$, or

$$S^2 = 2\,(\theta_1 - \theta_o)^{½} \int_{\theta_o}^{\theta_1} (\theta - \theta_o)^{½}\, D(\theta) d\theta. \quad [5]$$

*Extracted from: Parlange, J.-Y., 1975. Determination of Soil Water Diffusivity by Sorptivity Measurements. Soil Science Society of America Journal 39, 1011–1012. https://doi.org/10.2136/sssaj1975.03615995003900050057x*

Alternative formulations have been proposed for the definition of flux concentration functions $F(\theta)$ and corresponding expression for sorptivity by Angulo-Jaramillo et al. (2016):

**Table 1.1**  Main approximations of the flux-concentration function and sorptivity

| Reference | $F(\theta)$ | $S^2\ [L^2 T^{-1}]$ | Comment |
|---|---|---|---|
| Crank (1979) | $\exp\left\{ -\left[ inverfc\left( \frac{\theta - \theta_i}{\theta_0 - \theta_i} \right) \right]^2 \right\}$ | $2D \int_{\theta_i}^{\theta_0} \frac{\theta - \theta_i}{\exp\left\{ -\left[ inverfc\left( \frac{\theta - \theta_i}{\theta_0 - \theta_i} \right) \right]^2 \right\}}\, d\theta$ | Linear soil (i.e. constant diffusivity) |
| Philip and Knight (1974) | $\frac{\theta - \theta_i}{\theta_0 - \theta_i}$ | $2(\theta_0 - \theta_i) \int_{\theta_i}^{\theta_0} D(\theta)\, d\theta$ | Soil with Dirac δ-function diffusivity (i.e. Green and Ampt model) |
| Parlange (1975) | $\frac{2(\theta - \theta_i)}{\theta_0 + \theta - 2\theta_i}$ | $\int_{\theta_i}^{\theta_0} (\theta_0 + \theta - 2\theta_i) D(\theta)\, d\theta$ | Soils with strong non-linear behavior |
| Brutsaert (1976) | $\left( \frac{\theta - \theta_i}{\theta_0 - \theta_i} \right)^{1/2}$ | $2(\theta_0 - \theta_i)^{1/2} \int_{\theta_i}^{\theta_0} (\theta - \theta_i)^{1/2} D(\theta)\, d\theta$ | Soils with moderate non-linear behavior |

$\Theta = \frac{\theta - \theta_i}{\theta_0 - \theta_i}$, non-dimensional form of volumetric soil water content

*Extracted from: Angulo-Jaramillo, R., Bagarello, V., Iovino, M., Lassabatere, L., 2016. Infiltration measurements for soil hydraulic characterization, Infiltration Measurements for Soil Hydraulic Characterization. Springer, Switzerland. https://doi.org/10.1007/978-3-319-31788-5*

We rewrote this part in a clearer way in the revised version by developing the concept of flux-concentration function (see revised version, lines 25-37, marked revised manuscript).

L 21: initial and final water contents of the soil or the water source? More detailed information

about the relationship between sorptivity and water content is needed.

**Authors:** The Reviewer is correct. By final water content or water pressure head, the authors mean the conditions imposed at the surface (i.e., the water source). This is now detailed in the revised version of the manuscript (Lines 29-31, marked revised manuscript). We also added a section on the practical computation of sorptivity and its link to the initial and final water contents.

L 24: I couldn't find the same expression in Ross et al. (1996). Please give more details about the construction of this equation.

**Authors:** The Reviewer is correct. Ross et al. (1996) did not clearly write the equation as it is mentioned in our study. The two expressions, expressed as a function of $\theta$ or h, are equivalent as long as the upper boundary remains below $h_a$ (the air-entry pressure head). This equivalency comes directly from a change of variable $\theta \rightarrow h$. Both expressions give access to the unsaturated sorptivity. Otherwise, the expression of sorptivity should be expressed as a function of the water pressure head in order to include the saturated part of sorptivity. This approach was previously studied by Haverkamp et al. (1990).

Thus, irrespective of the $h(\theta)$ expression chosen, one may consider two branches for $h(\theta)$: one straight part for $h_{str} < h < 0$ with $\theta = \theta_s$ and one part given by one of the $h(\theta)$ equations mentioned before (e.g. Van Genuchten 1980). In consequence, the corresponding diffusivity function $D(\theta)$ has two components as well: a finite part $D_c(\theta)$, valid over the interval $[\theta_0, \theta_s]$, where the subscript $c$ refers to it being a continuous function, and an infinite part, $D_\Delta(\theta)$ for $\theta = \theta_s$, where the subscript $\delta$ refers to it being approached by the use of a standard delta function.

$$D(\theta) = D_c(\theta) + D_\Delta(\theta) \qquad (9)$$

or

$$D(\theta) = D_c(\theta) - K_s h_{str} \Delta(\theta - \theta_s), \qquad (10)$$

$K_s$ is the hydraulic conductivity at natural saturation $K(\theta_s)$, and $\Delta(\theta - \theta_s)$ is the delta function. Hence, $D(\theta)$ is infinite at $\theta = \theta_s$ but remains integrable.

The sorptivity, $S$, can be written similarly in a decomposed form by considering the limit of Eq. (8) as time goes to zero, or, for $z_s = 0$,

$$S^2 = S_c^2 - 2\,h_{str} K_s [\theta_s - \theta_0] \qquad (11)$$

with

$$S_c^2 = 2 \int_{\theta_0}^{\theta_s} \frac{[\theta - \theta_0]}{F(\theta,0)} D_c(\theta)d\theta \qquad (12)$$

We have changed this part in line with Lassabatere et al. (2021). In the new version, we simply state that in our previous paper, we suggested using both forms, expressed as a function of water content or water pressure head, and we then state that the second form was more general and included the saturated part of sorptivity. We rely our statement on the study by Haverkamp et al. (1990), previously reported by many authors (e.g., Stewart et al., 2013).

L 240-246: Please use these ideas to build hypothesis and objectives, and include them in the Introduction section.

**Authors:** The Reviewer is correct. This part was rewritten to state the objectives in a much clearer way at the end of the introduction.

L 363-371: this is not a conclusion. The inclusion of an explicit hypothesis will improve this section.

**Authors:** The Reviewer is correct. The conclusion was strengthened in the revised manuscript, with more details on the taking-home messages and more insight into perspectives and further

works.

**Anonymous Referee #2**

Referee comment on "Mixed formulation for an easy and robust numerical computation of sorptivity" by Laurent Lassabatere et al., Hydrol. Earth Syst. Sci. Discuss., https://doi.org/10.5194/hess-2021-633-RC2, 2022

Overall manuscript is well written. Authors need to include Novelty.

**Authors:** The authors thank the Reviewer for his review. Additional results were added, and the novelty of the paper was highlighted in a much more straightforward way in the introduction and the conclusion (lines 68-75, introduction of the marked revised manuscript, and lines 560-567, conclusions of the marked revised manuscript). To the authors' knowledge, no mixed formulation has been previously proposed, and the investigation of the impact of misestimating sorptivity on modeling water infiltration into soils has never been studied in depth.